# Measuring and Mitigating Identity Bias in Multi-Agent Debate via Anonymization

## Abstract

Multi-agent debate (MAD) aims to improve large language model (LLM) reasoning by letting multiple agents exchange answers and then aggregate their opinions. Yet recent studies reveal that agents are not neutral: they are prone to identity-driven sycophancy and self-bias, uncritically adopting a peer's view or stubbornly adhering to their own prior output, undermining the reliability of debate. In this work, we present the first principled framework that joins sycophancy and self-bias to mitigate and quantify identity bias in MAD. First, we formalize the debate dynamics as an identity-weighted Bayesian update process. Second, we propose response anonymization: by removing identity markers from prompts, agents cannot distinguish "self" from "peer", which forces equal weights on agent identity, thereby reducing bias. Third, we define the Identity Bias Coefficient (IBC), a principled metric that measures how often an agent follows a peer versus itself. Empirical studies across multiple models, datasets and debate rounds confirm that identity bias is widespread, with sycophancy far more common than self-bias. Our findings highlight the need to "mask" identity to ensure that MAD systems reason based on content rather than source identity.

## 1 Introduction

Humans have long relied on collective reasoning as a means of resolving uncertainty and reaching better decisions. Courtrooms, round tables, and scientific peer review all testify to the power of group decision-making. Drawing inspiration from these settings, the multi-agent debate (MAD) paradigm has been proposed as a method for strengthening the reasoning capabilities of large language models (LLMs) (Chan et al., 2024; Du et al., 2024; Bo et al., 2024; Li et al., 2024c). In a typical MAD system, several LLM agents are asked to solve a shared task, observe one another's responses, and iteratively revise their answers before a final aggregation step. The intended effect of this system is to reinforce correct reasoning signals and enable mutual error correction.

Yet, the reliability of MAD remains contested. A key—but underexplored—factor behind these failures lies in *identity-driven biases*: agents' tendency to respond differently depending on whether information originates from themselves or from their peers. Such biases can be categorized into two forms. Sycophancy occurs when an agent overweights peer responses, deferring even when its own beliefs are stronger. Self-bias, in contrast, arises when an agent disproportionately clings to its own prior outputs, ignoring valid counter-evidence. While both phenomena are well-documented in single-agent user interactions (Li et al., 2025b; Fanous et al., 2025; Liu et al., 2025b; Barkett et al., 2025; Malmqvist, 2025; Hong et al., 2025; Spiliopoulou et al., 2025; Chen et al., 2025c; Laurito et al., 2025; Chen et al., 2025b; Yuan et al., 2025), their role in shaping MAD dynamics has not been systematically investigated.

In this work, we first introduce a theoretical framework that rigorously models how agents' identity biases manifest within MAD dynamics. We show that identity bias can distort debate dynamics, leading to premature consensus and erosion of MAD's intended benefits. To capture these effects, we introduce interpretable metrics—*Conformity* and *Obstinacy*—which measure an agent's tendency to align with its peer's prior answer versus its own prior answer under disagreement. Building on a probabilistic formalization of debate, we model agents as sampling from latent belief distributions that are updated through peer interactions. Within this framework, we prove that the gap between Conformity and Obstinacy admits a clean decomposition into two terms: a belief difference

term, reflecting genuine content-driven asymmetries between self and peer, and an identity bias term, capturing distortions introduced solely by the labeling of responses as "self" or "peer." This decomposition provides a principled way to separate rational belief updating from identity-driven distortions. Importantly, it reveals that much of the skew observed in practice does not originate from the agent's belief state, but rather from asymmetries in how identities are weighted during the update process.

Motivated by our theory, we propose a simple yet powerful intervention: Response Anonymization. In standard debate prompts, each response is explicitly labeled by its source—whether it was generated by the agent itself or by a peer. These identity markers create the very channel through which sycophancy and self-bias arise. Anonymization removes this channel: by masking all identity labels from debate transcripts, the agent is presented with arguments without attribution. The key advantage of our method lies in its minimalism: it requires no model retraining, no auxiliary loss functions, and no architectural modifications. It is directly applicable across different model families and debate settings. At the same time, it preserves the substance of deliberation—agents still exchange and evaluate arguments—but eliminates the systematic distortions introduced by identity.

Extensive experiments across diverse models and benchmarks demonstrate both the pervasiveness of identity bias and the effectiveness of Response Anonymization in mitigating it. Notably, on MMLU, Qwen-32B (Yang et al., 2024) exhibits a large Conformity–Obstinacy gap (Sec. 4.1 Theorem 1) of 0.608 in the vanilla setting, which reduces to just 0.024 under anonymization—a nearly complete removal of identity-driven distortion. Similar reductions are observed across other models and tasks, confirming that anonymization is a lightweight yet consistently effective method for aligning MAD dynamics with their intended purpose. We summarize our contributions as follows:

1. We formalize the debate process as a Bayesian belief update that explicitly incorporates the influence of agent identities. Our framework captures both directions of identity-driven behavior: sycophancy and self-bias. To the best of our knowledge, this is the first work to unify these concepts under the notion of identity bias.
2. We propose *Response Anonymization*, a simple yet effective approach to preclude identity-driven bias in multi-agent debate systems.
3. Building on our framework, we propose the *Identity Bias Coefficient* (IBC), a principled metric that quantifies the level of identity bias. We further extend our analysis to heterogeneous agents and multiple-peer settings, offering deeper insights into how identity bias shapes and influences the dynamics of debate.

## 2 PRELIMINARIES

**Multi-Agent Debate.** MAD is a collaborative framework in which multiple LLM agents engage in structured interactions by iteratively exchanging opinions and responses on a given task (Bo et al., 2024; Du et al., 2024; Chan et al., 2024; Tang et al., 2024; Wu et al., 2024; Chen et al., 2024c). A common design choice in MAD is the simultaneous-talk protocol (Chan et al., 2024), where agents asynchronously generate opinions at each debate round and iteratively exchange them in a structured manner. At round $t$, each agent observes both its own and its designated peers' responses from round $t - 1$, then updates its output with respect to the context. After multiple rounds, a final decision is typically obtained via an aggregation mechanism—most often majority voting. The goal of MAD is to leverage the ensemble effect of diverse reasoning paths from multiple agents, while critically examining the validity of the peer opinions to improve the overall quality of the final answer.

**MAD Protocol Formalization.** Let $(\mathcal{X}, \mathcal{Y})$ denote the input and output spaces of an agent. Each agent is modeled as a stochastic function $\pi_i : \mathcal{X} \to \mathcal{Y}$, typically an LLM, where $i \in \{1, 2, \ldots, N\}$ indexes the agents participating in the multi-agent debate (MAD) system. At the initial round $t = 0$, each agent produces an answer $y_{i,0} \in \mathcal{Y}$ by sampling from $\pi_i(x)$ for a given input question $x \in \mathcal{X}$. At each subsequent debate round $t \geq 1$, agent $i$ observes the responses of its peers from the previous round: $Y_{i,t-1} = \{y_{j,t-1} \mid j \in \mathcal{P}(i)\}$, where $\mathcal{P}(i) \subseteq \{1, \ldots, N\}$ is the set of peers assigned to agent $i$. The agent may also optionally condition on its own prior output $y_{i,t-1}$, yielding the round-$t$ response:

$$y_{i,t} = \pi_i\big(x \,;\, Y_{i,t-1}, y_{i,t-1}\big).$$

After $T$ rounds, the system aggregates the final set of responses $\{y_{i,T}\}_{i=1}^{N}$ using majority voting to produce the debate outcome.

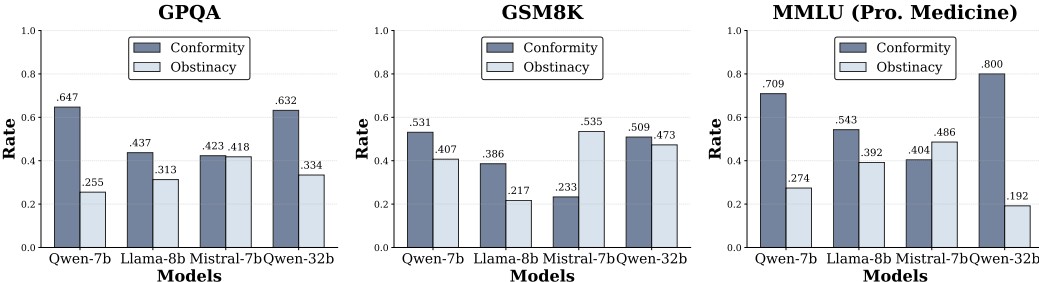

Figure 1: **Conformity vs. Obstinacy**. Comparison is done on a 5-agent MAD system with a single peer assigned to each agent. The versions of the four models are Qwen2.5-7b-instruct, Llama3.1-8b-instruct, Mistral-7b-instruct-v0.3, Qwen2.5-32b-instruct, respectively.

## 3 IS IDENTITY BIAS A PROBLEM IN MULTI-AGENT DEBATE?

In this section, we show that LLM agents engaged in multi-agent debate are susceptible to *identity-driven biases*, which distort the intended dynamics of collective reasoning. Two prominent extreme forms of identity bias are sycophancy and self-bias. Sycophancy refers to the tendency of an LLM to uncritically adopt the views or preferences of a peer agent or user, often at the expense of factual accuracy or principled reasoning (Li et al., 2025b; Fanous et al., 2025; Liu et al., 2025b; Barkett et al., 2025; Malmqvist, 2025; Hong et al., 2025). Self-bias, in contrast, occurs when an LLM disproportionately favors its own prior outputs over those of its peers, even when alternative responses may be more accurate or better reasoned (Spiliopoulou et al., 2025; Chen et al., 2025c; Laurito et al., 2025; Chen et al., 2025b; Yuan et al., 2025).

Prior studies have primarily investigated these biases in *single-agent user interactions*. However, *systematic analysis of identity bias in multi-agent debate remains scarce*. Our framework unifies sycophancy and self-bias under the broader notion of identity bias, emphasizing their impact on the dynamics of deliberation. Both forms of biases can undermine the core purpose of MAD—leading to premature consensus, reinforce incorrect responses, and weaken the reliability of aggregated outcomes. Understanding and mitigating these biases is therefore central to evaluating the reliability of MAD as a paradigm for reasoning and decision-making with LLMs.

### 3.1 MOTIVATING ANALYSIS

Here, we first introduce quantitative metrics that capture the behavioral tendencies of debate agents. Specifically, we define the **Conformity** and the **Obstinacy**, which measure, respectively, an agent's inclination to align with its peer versus to adhere to its own prior output. To ground the analysis in the simplest nontrivial interaction, we begin with the homogeneous single-peer setting: agents share the same base model architecture and persona, and each agent observes only one other agent (Chan et al., 2024; Du et al., 2024; Li et al., 2024c; Wang et al., 2024a; Zhang et al., 2024). This avoids confounding effects from group dynamics and provides a clean lens through which to study identity-driven behavior. Moreover, this setting is a sparse communication structure, which is practically useful because it is often reported to be superior to the fully-connected topology (Li et al., 2024c; Estornell & Liu, 2024; Zhang et al., 2024). Extension to the multi-peer setup will be discussed in Sec. 6.3. For agent $i$ with respect to its peer agent $j$, we define:

$$\text{Conformity}_i := \mathbb{E}[\mathbf{1}\{y_{i,t} = y_{j,t-1}\} \,|\, y_{i,t-1} \neq y_{j,t-1}] \tag{1}$$

$$\text{Obstinacy}_i := \mathbb{E}[\mathbf{1}\{y_{i,t} = y_{i,t-1}\} \,|\, y_{i,t-1} \neq y_{j,t-1}], \tag{2}$$

where $y_{i,t}$ and $y_{j,t}$ denote the answers produced by agents $i$ and $j$ ($i \neq j$) at round $t$. The *Conformity* captures the degree to which agent $i$ aligns with its peer's prior answer in the presence of disagreement, while the *Obstinacy* reflects its propensity to remain self-reliant by repeating its own prior answer. Together, these indices provide interpretable, task-level statistics that allow us to compare and contrast identity-driven behaviors across models and tasks.

**Empirical Findings.** In Figure 1, we compare the Conformity and Obstinacy metrics across four LLMs on three benchmark datasets. We take the aggregate statistic from $N = 5$ agents across multiple dataset samples to estimate them (see details in Appendix A.3). The gaps between

the two metrics are generally substantial, demonstrating that identity bias manifests to varying degrees across models and benchmarks. In most cases, Conformity exceeds Obstinacy, suggesting a dominant sycophantic tendency in LLM debate agents. Nevertheless, we also observe notable exceptions—such as Mistral-7B on GSM8K—where Obstinacy surpasses Conformity, suggesting that self-bias, though less frequent, can emerge as a significant factor in certain scenarios. These findings underscore the need for precise characterization of identity-driven behaviors, motivating the following section to formally model how identity bias influences debate dynamics and to introduce a method for eliminating its effects.

## 4 ELIMINATING IDENTITY BIAS VIA ANONYMIZATION

In this section, we introduce a theoretically grounded framework for quantifying and eliminating identity bias in multi-agent debate. We begin by formalizing debate dynamics as an identity-driven Bayesian belief update process. Then, we establish how the *Conformity* and *Obstinacy* map onto this update, thereby disentangling identity effects from belief-driven reasoning (Sec. 4.1). Finally, we propose a theoretically motivated intervention—*Response Anonymization*—as a simple and effective communication strategy to eliminate identity bias (Sec. 4.2).

### 4.1 FORMALIZING MULTI-AGENT DEBATE UNDER IDENTITY BIAS

To rigorously capture how individual agents generate responses within this debate framework, Choi et al. (2025) introduced a probabilistic modeling perspective. *However, prior work treats peer influence and self-reliance uniformly and does not consider identity bias in the modeling.* In contrast, our formalization explicitly distinguishes between two distinct behavioral tendencies: sycophancy (alignment with peers) and self-bias (persistence on one's own prior outputs). This allows us to capture systematic deviations from unbiased belief updating.

In this framework, an agent's behavior is formalized as arising from an underlying belief distribution over possible answers, and the belief update process is determined by its neighboring peer responses. This allows us to account for both the diversity of reasoning paths across agents and the stochasticity inherent in the MAD system. In particular, each agent is an idealized generative model governed by a Dirichlet-Compound-Multinomial (DCM) distribution. The Dirichlet prior captures the agent's internal belief over possible answers, while the Multinomial models the stochastic generation process (*e.g.*, via temperature or nucleus sampling). This distribution is thus a natural choice because it encapsulates both internal uncertainty and output randomness, while also providing a principled Bayesian framework for belief updates across debate rounds—enabling analytical study of dynamics during the debate process.

**Definition 1. (Agent Response Generation under DCM Model)** *Consider an agent $i$ at debate round $t$. The agent maintains a belief parameter vector $\boldsymbol{\alpha}_{i,t} = (\alpha_{i,t}^{(1)}, \ldots, \alpha_{i,t}^{(K)}) \in \mathbb{R}_+^K$, where each component $\alpha_{i,t}^{(k)}$ quantifies its confidence in option $k \in \mathcal{A}$. A response is produced through the following generative mechanism:*

$$\begin{aligned} \text{(Belief sampling)} \quad &\boldsymbol{\theta}_{i,t} \sim \text{Dirichlet}(\boldsymbol{\alpha}_{i,t}), \\ \text{(Response generation)} \quad &y_{i,t} \sim \text{Categorical}(\boldsymbol{\theta}_{i,t}). \end{aligned}$$

*Marginalizing over the Dirichlet sample $y_{i,t} \in \mathcal{A}$, the probability of choosing answer $k$ is expressed as $P(y_{i,t} = k \mid \boldsymbol{\alpha}_{i,t}) = \alpha_{i,t}^{(k)}/||\boldsymbol{\alpha}_{i,t}||_1$.*

Building on this definition, we will formalize how an agent's belief evolves throughout debate as a function of both its own prior response and those of its peers. We characterize this evolution with respect to the agent's preferential bias toward a specific identity.

**Identity-driven Belief Update.** To better understand the identity-driven behaviors of agents, it is useful to think of them as shaping the way agents update their beliefs during debate. Each response from an agent or its peers can be viewed as evidence, but sycophancy and self-bias change how this evidence is weighted. Instead of treating all responses equally, a sycophantic agent may place extra weight on peer opinions, while a self-biased agent may lean more heavily on its own prior outputs. For example, when two agents disagree, a sycophantic one might still copy its peer's answer despite

**Q.** Mary had 3 apples, but she ate 2 of them. How many apples are left?

Figure 2: **Response Anonymization.** By anonymizing the responses in multi-agent debate, an agent's answer is driven entirely by its belief state, rather than the agents' identity information.

having stronger initial confidence in its own, while a self-biased one might stubbornly reinforce its prior choice even in the face of clear counterevidence. By framing these behaviors as a Bayesian update with adjustable weights, we can capture such systematic tendencies in a transparent and analyzable way. This motivates the following definition of identity-driven Bayesian belief updates. Building upon the DCM model from Definition 1, we define:

**Definition 2. (Identity-driven Bayesian Belief Update from Agent Responses)** *Let* $\{y_{j,t-1} \mid j \in \mathcal{P}(i) \cup \{i\}\}$ *be the set of responses observable to agent* $i$ *from its peers* $\mathcal{P}(i)$ *at round* $t$. *These responses induce a count vector* $\boldsymbol{c}_{i,t} = w_i\, \boldsymbol{e}_{i,t} + \sum_{j \in \mathcal{P}(i)} w_j\, \boldsymbol{e}_{j,t}$, *where* $w_i, w_j > 0$ *are the identity weights, and* $\boldsymbol{e}_{i,t}, \boldsymbol{e}_{j,t} \in \mathbb{B}^K$ *are one-hot vectors indicating the answer chosen out of* $K$ *possible answers. Then, the agent updates its Dirichlet parameter as:* $\boldsymbol{\alpha}_{i,t} = \boldsymbol{\alpha}_{i,t-1} + \boldsymbol{c}_{i,t}$.

Definition 2 defines that the way agents incorporate evidence during debate is not only a matter of content but also of identity. By allowing different weights on self versus peer responses, the update rule makes explicit how sycophancy or self-bias can systematically distort the belief evolution of an agent. This has important implications: identity bias can amplify errors by overweighting unreliable sources, or suppress corrective signals that would otherwise arise from diverse perspectives. At the same time, the weighted formulation provides a handle for analyzing and mitigating such behaviors, since interventions can target the relative weighting scheme rather than the entire belief update process. Based on the DCM model, we can provide a closed-form expression for the measurements:

**Theorem 1. (Conformity and Obstinacy under Identity-Driven Updates)** *Consider agent* $i$ *and its peer* $j$ *in the identity-driven Bayesian belief update model (Definition 2), where* $y_{i,t-1} \neq y_{j,t-1}$. *Let* $\alpha_{i,t-1}^{(k)}$ *denote agent* $i$'s *belief mass on answer* $k$ *at round* $t-1$, *and let* $w_i, w_j > 0$ *be the identity weights for self and peer, respectively. Then, the Conformity and Obstinacy defined in Sec. 3.1 can be expressed as*

$$Conformity_i = \frac{\alpha_{i,t-1}^{(y_{j,t-1})} + w_j}{\|\boldsymbol{\alpha}_{i,t}\|_1}, \qquad Obstinacy_i = \frac{\alpha_{i,t-1}^{(y_{i,t-1})} + w_i}{\|\boldsymbol{\alpha}_{i,t}\|_1}. \tag{3}$$

*Moreover, their difference admits the decomposition*

$$\Delta_i := Conformity_i - Obstinacy_i = \frac{1}{\|\boldsymbol{\alpha}_{i,t}\|_1} \left( \underbrace{(\alpha_{i,t-1}^{(y_{j,t-1})} - \alpha_{i,t-1}^{(y_{i,t-1})})}_{\text{belief difference}} + \underbrace{(w_j - w_i)}_{\text{identity bias}} \right). \tag{4}$$

*Proof.* See Appendix C.1 for proof, Appendix C.3 for parameter estimation, and Sec. 6.3 for multi-peer extensions.

This form of expression reveals that conformity is governed jointly by the agent's prior belief in its peer's answer and the corresponding identity weight, while obstinacy is analogously determined by its prior belief in its own answer and its self-weight. The quantity $\Delta_i$ provides a direct measure of agent $i$'s relative orientation toward its peer versus itself. It is jointly determined by two components: (i) the *belief difference*, capturing the relative prior confidence in the peer's answer versus the agent's own, and (ii) the *identity bias*, capturing the asymmetry in how identity is weighted during the belief update. In the ideal case, the identity bias term vanishes (*i.e.*, $w_j = w_i$), so that the agent's decisions depend exclusively on its underlying belief state. Guided by the theory, the next section introduces an approach for eliminating this identity bias through response anonymization.

## 4.2 RESPONSE ANONYMIZATION

The decomposition in Theorem 1 reveals that an agent's relative orientation toward its peer versus itself, $\Delta_i$, is shaped not only by differences in prior beliefs but also by asymmetries in how identity is weighted. This leads to the following immediate consequence:

**Corollary 1. (Absence of Identity Bias)** *If the identity weights are symmetric, i.e. $w_i = w_j$ for $j \in \mathcal{P}(i)$, then the difference between Conformity and Obstinacy reduces to*

$$\Delta_i = \frac{\alpha_{i,t-1}^{(y_{j,t-1})} - \alpha_{i,t-1}^{(y_{i,t-1})}}{\|\boldsymbol{\alpha}_{i,t}\|_1}.$$

*In this case, the relative tendency of agent $i$ to conform versus remain obstinate depends solely on its prior belief distribution, independent of identity-driven effects.*

Corollary 1 suggests a natural design principle: if we can enforce symmetry in identity weights, the influence of identity bias disappears and agents behave according to their beliefs alone. Standard debate prompts (Appendix B.1), however, explicitly disclose the identity of each response, allowing the agent to condition its update on whether an answer came from itself or from a peer. This disclosure provides the very channel through which identity bias can arise. Our intervention is to *anonymize* the prompt by removing all identity markers (Appendix B.2). In the anonymized setting, the agent is presented with responses without attribution, and thus has no basis for assigning different weights to self versus peer. This symmetry enforces equal identity weights, $w_i = w_j$, and thereby eliminates any systematic preference for "self" or "peer" labels. In other words, after anonymization, the agent's relative tendency to align with its peer versus itself is driven entirely by its belief state $\boldsymbol{\alpha}_{i,t-1}$, rather than by identity information. This ensures that any residual bias reflects content-based evaluation rather than identity-driven sycophancy or self-bias. A visual overview of this anonymization process is provided in Figure 2.

**Identity Bias Coefficient (IBC).** To directly quantify the role of identity asymmetry in shaping agent behavior, we define the *Identity Bias Coefficient* (IBC):

$$\text{IBC}_i = \Delta_i^{\text{vanilla}} - \Delta_i^{\text{anonymized}} = \frac{w_j - w_i}{\|\boldsymbol{\alpha}_{i,t}\|_1}. \tag{5}$$

This metric captures the portion of $\Delta_i$ attributable *solely* to identity bias, after removing the influence of belief differences. In other words, $\text{IBC}_i$ measures how much agent $i$'s relative orientation toward its peer versus itself is shifted by identity labels. A positive IBC indicates a stronger weighting of the peer's identity (*sycophancy*), while a negative IBC indicates a stronger weighting of the agent's own identity (*self-bias*).

## 5 EXPERIMENTS

### 5.1 SETUP

**Models and Datasets.** We evaluate across five model families: `Qwen2.5-7b-instruct`, `Qwen2.5-32b-instruct` (Yang et al., 2024), `Llama3.1-8b-instruct` (Grattafiori et al., 2024), `Mistral-7b-v0.3` (Jiang et al., 2023), and latest `GPT-OSS-20b` (Agarwal et al., 2025), and evaluate on four benchmark datasets covering diverse reasoning tasks: Google-Proof QA (GPQA) (Rein et al., 2024), MMLU Professional Medicine subset (Hendrycks et al., 2021b;a), HellaSwag (Zellers et al., 2019), and the Grade-School Math 8K (GSM8K) (Cobbe et al., 2021). See Appendix A.1 for more dataset details, and Appendix A.2 for other experimental details.

### 5.2 EXPERIMENTAL RESULTS

**Response anonymization reduces identity bias.** As shown in Table 1, the $\Delta$ values under the base agent setting often exhibited substantial magnitudes, roughly capturing the presence of identity bias across different model families and datasets. For example, on MMLU, Qwen-32B shows $\Delta = 0.608$ in the vanilla setting. After applying Response Anonymization, this value drops to $\Delta = 0.024$, confirming that much of the original effect was attributable to identity bias. Similar reductions are observed across other models and benchmarks, highlighting the general effectiveness

Table 1: **Effects of Response Anonymization on Identity Bias.** ✗ and ✓ are the base agent and the response-anonymized agent measurements, respectively. The positive Identity Bias Coefficients are colored blue, and red for negative values. The highlighted 'IBC' row shows the value difference between the top two rows. We retrieved the measurements from the first round of debate.

| Agent | Anonymize | GPQA | | | MMLU (Pro. Medicine) | | | HellaSwag | | | GSM8K | | |
|---|---|---|---|---|---|---|---|---|---|---|---|---|---|
| | | Conf. | Obst. | Δ | Conf. | Obst. | Δ | Conf. | Obst. | Δ | Conf. | Obst. | Δ |
| Llama-8B | ✗ | 0.437 | 0.313 | 0.124 | 0.543 | 0.392 | 0.151 | 0.569 | 0.308 | 0.261 | 0.386 | 0.217 | 0.169 |
| | ✓ | 0.389 | 0.363 | 0.026 | 0.392 | 0.549 | -0.157 | 0.465 | 0.456 | 0.009 | 0.406 | 0.317 | 0.089 |
| | IBC | | | ↓ 0.098 | | | ↓ 0.307 | | | ↓ 0.252 | | | ↓ 0.080 |
| Mistral-7B | ✗ | 0.423 | 0.418 | 0.005 | 0.404 | 0.486 | -0.082 | 0.485 | 0.449 | 0.036 | 0.233 | 0.535 | -0.302 |
| | ✓ | 0.378 | 0.460 | -0.082 | 0.408 | 0.475 | -0.067 | 0.428 | 0.492 | -0.064 | 0.302 | 0.459 | -0.157 |
| | IBC | | | ↓ 0.087 | | | ↑ -0.015 | | | ↓ 0.100 | | | ↑ -0.145 |
| Qwen-7B | ✗ | 0.647 | 0.255 | 0.392 | 0.709 | 0.274 | 0.435 | 0.747 | 0.240 | 0.507 | 0.531 | 0.407 | 0.124 |
| | ✓ | 0.485 | 0.424 | 0.061 | 0.498 | 0.471 | 0.027 | 0.484 | 0.516 | -0.032 | 0.414 | 0.510 | -0.096 |
| | IBC | | | ↓ 0.331 | | | ↓ 0.408 | | | ↓ 0.539 | | | ↓ 0.220 |
| Qwen-32B | ✗ | 0.632 | 0.334 | 0.298 | 0.800 | 0.192 | 0.608 | 0.696 | 0.304 | 0.392 | 0.509 | 0.473 | 0.036 |
| | ✓ | 0.502 | 0.466 | 0.036 | 0.512 | 0.488 | 0.024 | 0.536 | 0.455 | 0.081 | 0.455 | 0.509 | -0.054 |
| | IBC | | | ↓ 0.262 | | | ↓ 0.584 | | | ↓ 0.311 | | | ↓ 0.092 |
| GPT-OSS-20B | ✗ | 0.359 | 0.319 | 0.040 | 0.618 | 0.382 | 0.236 | 0.588 | 0.408 | 0.180 | 0.568 | 0.378 | 0.190 |
| | ✓ | 0.335 | 0.371 | -0.036 | 0.509 | 0.473 | 0.036 | 0.460 | 0.529 | -0.069 | 0.528 | 0.417 | 0.111 |
| | IBC | | | ↓ 0.076 | | | ↓ 0.200 | | | ↓ 0.249 | | | ↓ 0.079 |

of anonymization as a mitigation strategy. In a homogeneous two-agent setting, the expected value of $\Delta_i^{\text{anonymized}}$ is zero because, for the pair of agents 1 and 2, their belief-difference terms satisfy $\Delta_1^{\text{anonymized}} = -\Delta_2^{\text{anonymized}}$. Nonetheless, the empirical estimates need not be exactly zero, as they naturally reflect variance arising from sample-level belief differences. The IBC removes this residual variance, allowing us to isolate the pure effect of identity bias independent of belief-differences.

**Sycophancy is more prevalent compared to self-bias in MAD.** Table 1 reports the Identity Bias Coefficient (IBC) values across models and datasets, which correspond to the quantities colored in blue and pink. As established in Sec. 4.2, the sign of IBC directly reflects whether an agent exhibits sycophantic (IBC > 0) or self-biased (IBC < 0) behavior. Out of 20 evaluated cases, 18 yield positive IBC values, while only 2 exhibit negative values. This clear skew toward positive values demonstrates that sycophancy is far more prevalent than self-bias in multi-agent debate.

**The level of identity bias varies across tasks and model families.** Although sycophancy emerges as the predominant pattern in our experiments, the magnitude of the Identity Bias Coefficient (IBC) is far from uniform across tasks or model families. For instance, Mistral-7B in Table 1 exhibits small IBC values, suggesting that it is comparatively less prone to identity-driven influence. Moreover, the relative scale of IBC differs substantially across benchmarks, highlighting that the degree of identity bias is task-dependent as well as model-dependent.

## 6 EXTENDED ANALYSES

### 6.1 IMPROVED TRUSTWORTHINESS

The core contribution of our response anonymization is improvement in trustworthiness of the MAD system. To concretely analyze the trustworthiness using two behavioral ratios, Subversion and Correction, defined as:

$$\text{Subversion} = \mathbb{P}\left[y_{i,t} = \text{incorrect} \mid y_{i,t-1} = \text{correct},\ y_{j,t-1} = \text{incorrect}\right] \quad (6)$$

$$\text{Correction} = \mathbb{P}\left[y_{i,t} = \text{correct} \mid y_{i,t-1} = \text{incorrect},\ y_{j,t-1} = \text{correct}\right]. \quad (7)$$

By comparing these ratios before and after anonymization in Table 2, we observe that the Subversion ratio consistently exhibits a larger relative drop than the Correction ratio. For instance, on the Professional Medicine (MMLU) benchmark with Qwen-32B, the Subversion ratio decreases by 64.3%, whereas the Correction ratio decreases by only 14.9% after anonymization. This indicates that LLM agents are more prone to subverting their originally correct answers when identities are visible, and that Identity Anonymization effectively reduces such undesirable behavior. However, despite the larger proportional drop in Subversion, we find that its overall positive effect on total accuracy is mitigated by the much larger number of Correction events. In other words, even though Subversion becomes significantly less frequent in ratio, the net accuracy impact is dominated by the greater volume of Correction cases, partially counteracting the benefit. Direct analysis on the performance is deferred to Appendix D.

Table 2: **Trustworthiness Improvement after Response Anonymization**.

| Agent | | GPQA | Pro. Medicine | HellaSwag | GSM8K |
|---|---|---|---|---|---|
| **Llama-8B** | Subv. (Drop %) | $0.615 \rightarrow 0.545$ (11.4%) | $0.507 \rightarrow 0.321$ (36.7%) | $0.632 \rightarrow 0.523$ (17.2%) | $0.513 \rightarrow 0.412$ (19.7%) |
| | Corr. (Drop %) | $0.566 \rightarrow 0.503$ (11.1%) | $0.649 \rightarrow 0.537$ (17.2%) | $0.637 \rightarrow 0.549$ (13.8%) | $0.481 \rightarrow 0.503$ (-4.6%) |
| **Mistral-7B** | Subv. (Drop %) | $0.528 \rightarrow 0.454$ (14.0%) | $0.381 \rightarrow 0.376$ (1.3%) | $0.541 \rightarrow 0.500$ (7.6%) | $0.494 \rightarrow 0.545$ (-10.3%) |
| | Corr. (Drop %) | $0.472 \rightarrow 0.435$ (7.8%) | $0.552 \rightarrow 0.543$ (1.6%) | $0.512 \rightarrow 0.436$ (14.8%) | $0.278 \rightarrow 0.295$ (-6.1%) |
| **Qwen-7B** | Subv. (Drop %) | $0.717 \rightarrow 0.500$ (30.3%) | $0.579 \rightarrow 0.389$ (32.8%) | $0.709 \rightarrow 0.430$ (39.4%) | $0.342 \rightarrow 0.233$ (31.9%) |
| | Corr. (Drop %) | $0.711 \rightarrow 0.553$ (22.2%) | $0.853 \rightarrow 0.632$ (25.9%) | $0.767 \rightarrow 0.488$ (36.4%) | $0.740 \rightarrow 0.575$ (22.3%) |
| **Qwen-32B** | Subv. (Drop %) | $0.473 \rightarrow 0.357$ (24.5%) | $0.750 \rightarrow 0.268$ (64.3%) | $0.630 \rightarrow 0.500$ (20.6%) | $0.455 \rightarrow 0.333$ (26.8%) |
| | Corr. (Drop %) | $0.736 \rightarrow 0.651$ (11.5%) | $0.839 \rightarrow 0.714$ (14.9%) | $0.739 \rightarrow 0.543$ (26.5%) | $0.727 \rightarrow 0.727$ (0.0%) |
| **GPT-OSS-20B** | Subv. (Drop %) | $0.164 \rightarrow 0.091$ (44.5%) | $0.064 \rightarrow 0.059$ (7.8%) | $0.573 \rightarrow 0.462$ (19.4%) | $0.397 \rightarrow 0.288$ (27.5%) |
| | Corr. (Drop %) | $0.882 \rightarrow 0.864$ (2.0%) | $0.965 \rightarrow 0.951$ (1.5%) | $0.581 \rightarrow 0.487$ (16.2%) | $0.809 \rightarrow 0.753$ (6.9%) |

## 6.2 HETEROGENEOUS AGENTS

Our exploration has thus far focused on MAD systems with homogeneous agents, where all participants share the same model architecture and persona. Then, a natural question arises: does identity bias persist at the same level when agents are heterogeneous? To investigate this, we evaluate identity bias metrics in MAD systems composed of agents with distinct personas. Following Liu et al. (2024b), we apply the persona set tailored for "clinical knowledge" tasks to

Table 3: **Heterogeneous Agents.**

| Agent | Persona | $\Delta$ vanilla | $\Delta$ w. anony | IBC |
|---|---|---|---|---|
| Qwen-7B | homogeneous | 0.435 | 0.027 | 0.408 |
| | heterogeneous | 0.457 | 0.083 | 0.374 |
| Qwen-32B | homogeneous | 0.608 | 0.024 | 0.584 |
| | heterogeneous | 0.445 | 0.055 | 0.390 |
| GPT-OSS-20B | homogeneous | 0.236 | 0.036 | 0.200 |
| | heterogeneous | 0.193 | 0.071 | 0.122 |

solve MMLU (Professional Medicine). The set includes a general-purpose "Assistant" as well as specialized roles such as "Doctor," "Psychologist," "Mathematician," and "Programmer." Each agent is initialized with a system prompt specifying its assigned role, using the same templates provided in Liu et al. (2024b) (see Appendix B.3 for the prompts).

Table 3 reports the comparison between homogeneous and heterogeneous configurations across three model families. Our results reveal two takeaways: (1) Response anonymization reliably eliminates identity-driven bias, even in the heterogeneous setting. For Qwen-7B, the raw $\Delta$ in the heterogeneous setting is $0.457$ without anonymization, but drops sharply to $0.083$ after anonymization—showing that much of the conformity–obstinacy gap vanishes once identity cues are removed. Similar trends hold across other models. (2) The IBC decreases when moving from homogeneous to heterogeneous agents (e.g., from $0.408$ to $0.374$ on Qwen-7B), suggesting that persona diversity reduces the extent to which behavior is driven by identity asymmetries.

## 6.3 EXTENSION TO MULTIPLE PEERS

While the single-peer setup is useful for isolating the effect of identity bias, practical MAD systems typically involve agents interacting with multiple peers simultaneously. We therefore extend the identity-driven belief update framework from Sec. 4.1 to a multi-peer setting.

**Formulation.** Given agent $i$'s peer set $\mathcal{P}(i)$, let $\mathcal{D}(i) := \{j \in \mathcal{P}(i) \mid y_{j,t-1} \neq y_{i,t-1}\}$ denote the set of peers that *disagreed* in the previous round, and $\mathcal{A}(i) := \{j \in \mathcal{P}(i) \mid y_{j,t-1} = y_{i,t-1}\}$ denote the ones that *agreed*. Also define $Y_{\mathcal{D}(i)} := \{y_{j,t-1} \mid j \in \mathcal{D}(i)\}$ as the set of peer answers that disagreed with agent $i$'s previous answer. Then, we generalize the Conformity and Obstinacy indices as follows:

$$\text{Conformity}_i := \mathbb{E}\left[\bigvee_{j \in \mathcal{D}(i)} \mathbf{1}\{y_{i,t} = y_{j,t-1}\} \,\middle|\, |\mathcal{D}(i)| = n_{\mathcal{D}} \neq 0, |\mathcal{A}(i)| = n_{\mathcal{A}}\right]$$

$$\text{Obstinacy}_i := \mathbb{E}[\mathbf{1}\{y_{i,t} = y_{i,t-1}\} \mid |\mathcal{D}(i)| = n_{\mathcal{D}} \neq 0, |\mathcal{A}(i)| = n_{\mathcal{A}}].$$

In this formulation, Conformity measures the probability that agent $i$ aligns with a disagreeing peer, while Obstinacy measures the probability that agent $i$ maintains its own prior response in the presence of $n_{\mathcal{D}}$ disagreeing peer agents.

Then, under Definition 2, the Dirichlet parameter update for agent $i$ is: $\boldsymbol{\alpha}_{i,t} = \boldsymbol{\alpha}_{i,t-1} + w_i \mathbf{e}_{i,t} + W_{\mathcal{A}} \mathbf{e}_{i,t} + \sum_{k \in Y_{\mathcal{D}(i)}} W^{(k)} \mathbf{e}^{(k)}$, where $W^{(k)} := \sum_{j \in \mathcal{P}(i)} w_j \mathbf{1}\{y_{j,t-1} = k\}$ is the aggregate peer weight for answer $k$, $W_{\mathcal{A}} := W^{(y_{i,t-1})} = \sum_{j \in \mathcal{A}(i)} w_j$, and $\mathbf{e}^{(k)}$ refers to the one-hot vector

representing answer $k$. This yields the following expressions for the indices:

$$\text{Conformity}_i := \frac{\sum_{k \in Y_{\mathcal{D}(i)}} \left( \alpha_{i,t-1}^{(k)} + W^{(k)} \right)}{||\boldsymbol{\alpha}_{i,t}||_1}, \qquad \text{Obstinacy}_i := \frac{\alpha_{i,t-1}^{(y_{i,t-1})} + w_i + W_{\mathcal{A}}}{||\boldsymbol{\alpha}_{i,t}||_1}.$$

The difference of the two indices can then be written as

$$\Delta_i := \frac{1}{||\boldsymbol{\alpha}_{i,t}||_1} \left( \underbrace{\sum_{k \in Y_{\mathcal{D}(i)}} \alpha_{i,t-1}^{(k)} - \alpha_{i,t-1}^{(y_{i,t-1})}}_{\text{belief difference}} + \underbrace{\sum_{k \in Y_{\mathcal{D}(i)}} W^{(k)} - w_i - W_{\mathcal{A}}}_{\text{identity-driven bias}} \right),$$

which parallels the structure of the single-peer case (equation 4). See Appendix C.2 for derivations.

If we assume homogeneous agents with $w_j \equiv w$, with $n_k := \sum_{j \in \mathcal{P}(i)} \mathbf{1}\{y_{j,t-1} = k\}$, each aggregate weight is $W^{(k)} = w\, n_k$ and $W_{\mathcal{A}} = w\, n_{\mathcal{A}}$. Then, the bias term reduces to:

$$\sum_{k \in Y_{\mathcal{D}(i)}} W^{(k)} - (w_i + W_{\mathcal{A}}) = (n_{\mathcal{D}} - n_{\mathcal{A}})\, w - w_i.$$

This incorporates the *bandwagon bias* (Ye et al., 2025): as the number of disagreeing peers increases, the aggregate peer influence grows proportionally, while its effect may be mitigated by the number of agreeing peers, $n_{\mathcal{A}}$. The single-peer case in equation 4 is recovered when $n_{\mathcal{D}} = 1$, $n_{\mathcal{A}} = 0$.

**Comparative Experiments.** We investigate the impact of peer group size on identity bias by comparing IBC values between single-peer and multi-peer ($|n_{\mathcal{D}}| = 4$) debate setups on Qwen-7B (Figure 3). Following the single-peer formulation, IBC is computed as the difference of $\Delta$ values derived from base and anonymized debates, respectively. Across all benchmarks, introducing multiple peers consistently reduces IBC, though the magnitude of change varies by task. These results suggest that the identity bias term is not a static property of the model, but a context-dependent value that is shaped by factors such as peer group size or answer quality. More discussion on relevant future directions is in Appendix F.

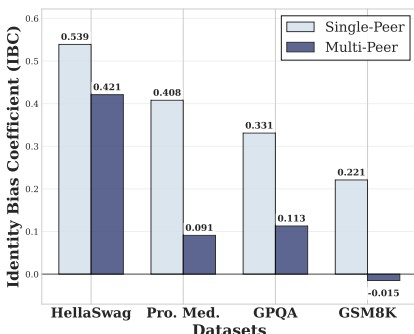

Figure 3: IBC drops in multi-peer setups.

## 7 RELATED WORKS

**Multi-Agent Debate.** Recently, there has been growing interest in multi-agent systems (MAS), with several surveys reviewing state-of-the-art LLM-based approaches (Guo et al., 2024; Tran et al., 2025; Yan et al., 2025; Li et al., 2024b). Within MAS, multi-agent debate has emerged as a promising paradigm for improving factual accuracy and reasoning in single-agent benchmarks, inspiring a range of task-specific applications (Bo et al., 2024; Du et al., 2024; Chan et al., 2024; Tang et al., 2024; Wu et al., 2024; Chen et al., 2024c), theoretical and protocol-level enhancements (Xiong et al., 2023; Li et al., 2024a; Chan et al., 2024; Liu et al., 2024a;b; Li et al., 2024c; Pham et al., 2024; Zhang et al., 2024), and strategies for encouraging diversity across agents (Chen et al., 2024a; Liu et al., 2024b; Liang et al., 2024; Wang et al., 2024b; Liu et al., 2025c; Chu et al., 2024) as well as learning-based methods to optimize debate dynamics (Liu et al., 2024b; Estornell et al., 2025; Chen et al., 2024d). Despite these advances, recent analyses have raised concerns about MAD's effectiveness: studies have documented numerous failure modes (Cemri et al., 2025), found that MAD does not consistently outperform single agents (Choi et al., 2025; Zhang et al., 2025a; Huang et al., 2024; Smit et al., 2024; Wang et al., 2024a), and highlighted tendencies toward incorrect answers (Xiong et al., 2023; Zhang et al., 2025a), majority-driven convergence (Estornell & Liu, 2024), or performance degradation with multiple rounds (Benedikt Kaesberg et al., 2025). Different from previous works, we *systematically examine the effect of identity bias and eliminate it via response anonymization*, thereby guiding the design of more reliable MAD systems.

**Sycophancy and Self-Bias.** Identity-driven biases in LLMs–notably sycophancy and self-bias–have been widely studied, though primarily in the context of single-agent user interactions. Prior work has analyzed sycophantic tendencies, where models uncritically align with external views (Sharma et al., 2024; Li et al., 2025b; Fanous et al., 2025; Liu et al., 2025b; Barkett et al., 2025; Malmqvist, 2025; Hong et al., 2025), and explored mitigation strategies (Wei et al., 2023; Rrv et al., 2024; Khan et al., 2024; Chen et al., 2024b; Zhang et al., 2025b). Related studies extend this line of inquiry to multi-modal models (Zhao et al., 2024; Li et al., 2025a), uncertainty quantification (Sicilia et al., 2025), and effect of assigning personas or roles for debates (Liu et al., 2025a; Bozdag et al., 2025; Chen et al., 2025a; Sandwar et al., 2025; Hu et al., 2025). In parallel, another body of work reports self-reliant behavior in LLMs–where models overly adhere to their own prior outputs (Wataoka et al., 2024; Panickssery et al., 2024; Davidson et al., 2024; Xu et al., 2024; Spiliopoulou et al., 2025; Chen et al., 2025c; Laurito et al., 2025)–with mitigation strategies also being investigated (Chen et al., 2025b; Yuan et al., 2025). However, discussions of identity bias in MAD remain scarce, with only a few works addressing sycophancy in this setup (Agarwal & Khanna, 2025; Pitre et al., 2025). In contrast, our work is, to the best of our knowledge, *the first to unify these two phenomena under the broader notion of "identity bias", and to propose a method that eliminates it from multi-agent systems.*

## 8    CONCLUSION

This work showed that LLM-based multi-agent debate systems are vulnerable to identity-driven biases: agents either defer to peers or cling to their own prior answers, undermining debate's goals of error correction and diverse reasoning. We unify these behaviors under an identity bias framework and model debate dynamics with a Bayesian update that incorporates agent identities. To mitigate bias, we proposed response anonymization, which removes identity markers and forces agents to weigh self and peer responses equally. Experiments across models and benchmarks reveal widespread, persistent identity bias, and that our proposed response anonymization can effectively eliminate it. Our framework provides both diagnostic and corrective tools, emphasizing that reliable MAD requires agents to reason from content rather than identity.

## ETHICS STATEMENT

This work aims to improve the reliability of multi-agent debate systems. We respect scientific integrity by presenting transparent theoretical derivations and rigorously evaluated metrics—Identity Bias Coefficient, Conformity, and Obstinacy—that quantify identity-driven biases. Our proposed response anonymization strategy is low-risk: it does not manipulate sensitive data or individuals, nor does it negatively impact privacy or welfare. We affirm that our interventions respect model neutrality and do not discriminate against any demographic group. All experimental setups use publicly available benchmarks. There are no conflicts of interest, and no human subjects were involved in data collection or evaluation.

## REPRODUCIBILITY STATEMENT

We have taken several steps to ensure the reproducibility of our work. All theoretical results are stated with full assumptions and complete proofs in the Appendix. Our experimental design is described in Section 5.1, with dataset details and preprocessing steps provided in Appendix A.1. Hyperparameter choices (temperature, nucleus sampling probability, maximum tokens) are reported in Appendix A.2, and all evaluations are conducted on publicly available benchmarks. We disclose the prompt templates in Appendix B as well. To further support reproducibility, we publicly release code and prompts. These resources enable independent verification of both our theoretical claims and empirical findings.

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

# Appendix

## Table of Contents

## A    EXPERIMENTAL DETAILS

### A.1    DATASET DETAILS

We provide dataset details and what portion of the data we used for our experiments.

**GPQA** (Rein et al., 2024) contains very difficult multiple-choice questions, written and verified by experts in the biology, physics, and chemistry domain. In particular, we use the 198 samples from the "Diamond" subset, which consists of high-quality samples that two experts answer correctly but most of the non-experts answer incorrectly.

**GSM8K** (Cobbe et al., 2021) comprises high-quality grade school math questions to evaluate the mathematical multi-step reasoning capabilities. We randomly select 300 samples from the original test split for our evaluations.

**MMLU (Professional Medicine)** (Hendrycks et al., 2021b;a) is a benchmark designed to evaluate professional-level reasoning in medical domains. It requires knowledge of medical concepts, clinical reasoning, and biomedical science to answer its questions. We use the full test split, which contains 272 items.

**HellaSwag** (Zellers et al., 2019) is a natural language inference (NLI) benchmark dataset focused on sentence completion. It evaluates whether a model can select the most plausible continuation of a given context from multiple candidates, a task requiring both linguistic competence and commonsense reasoning. From the original test split, we randomly sample 300 questions for our evaluations.

## A.2 IMPLEMENTATION DETAILS

**Hyperparameters.** We enable stochastic decoding by setting the sampling temperature to 1.0 and applying nucleus sampling with $p = 0.9$, restricting sampling to the dynamic set of tokens that together cover 90% of the probability mass. For all models, we generate up to 2048 tokens per response, to allow sufficient room for detailed reasoning.

**Resources.** All experiments were conducted using NVIDIA L40S, except for the experiments on GPT-OSS-20B that were done on Nvidia H200 GPUs.

## A.3 EVALUATION DETAILS

To capture population-level trends, we estimate Conformity and Obstinacy by averaging across $M$ dataset instances and $N$ agents:

$$\widehat{\text{Conformity}} := \frac{\sum_{m=1}^{M} \sum_{i=1}^{N} \mathbf{1}\{y_{i,t}^{(m)} = y_{j,t-1}^{(m)}\} \cdot \mathbf{1}\{y_{i,t-1}^{(m)} \neq y_{j,t-1}^{(m)}\}}{\sum_{m=1}^{M} \sum_{i=1}^{N} \mathbf{1}\{y_{i,t-1}^{(m)} \neq y_{j,t-1}^{(m)}\}},$$

$$\widehat{\text{Obstinacy}} := \frac{\sum_{m=1}^{M} \sum_{i=1}^{N} \mathbf{1}\{y_{i,t}^{(m)} = y_{i,t-1}^{(m)}\} \cdot \mathbf{1}\{y_{i,t-1}^{(m)} \neq y_{j,t-1}^{(m)}\}}{\sum_{m=1}^{M} \sum_{i=1}^{N} \mathbf{1}\{y_{i,t-1}^{(m)} \neq y_{j,t-1}^{(m)}\}}.$$

These estimates correspond to the maximum-likelihood estimators of the underlying conformity and obstinacy probabilities, justified obtained under the assumption of agent homogeneity and the i.i.d. nature of dataset samples. Given the estimations for these two root indices, we subsequently derive $\Delta$, $\tilde{\Delta}$, and the Identity Bias Coefficient (IBC), in our experiments.

# B PROMPT TEMPLATES

## B.1 STANDARD DEBATE PROMPT

The following is the standard debate prompt with two agents involved in the MAD system for a multiple-choice question task.

<question>

This was your most recent opinion:

- `<agent's response from the previous round>`

Based on the following other agents' opinions:

- Agent Opinion 1: `<peer agent's response from the previous round>`

Instructions: Consider these agents' opinions to provide an updated response to the question.

First, briefly state your step-by-step reasoning. Then, make sure to state your final answer in curly brackets at the very end of your response, just like: "{final answer: (A)}".

## B.2 ANONYMIZED DEBATE PROMPT

The following is the anonymized version of the debate prompt. Note that the order of the agent's responses presented is randomly determined.

<question>

Based on the following opinions from agents:

- Agent Opinion 1: `<an agent's response from the previous round>`

- Agent Opinion 2: `<an agent's response from the previous round>`

Instructions: Consider these agents' opinions to provide an updated response to the question.

First, briefly state your step-by-step reasoning. Then, make sure to state your final answer in curly brackets at the very end of your response, just like: "{final answer: (A)}".

### B.3 PERSONA PROMPTS

A persona-specific system prompt is assigned to each agent to allow heterogeneity. We adopt the persona prompts for "clinical knowledge", taken from Liu et al. (2024b), which are listed below:

- Assistant: You are a super-intelligent AI assistant capable of performing tasks more effectively than humans.

- Doctor: You are a doctor and come up with creative treatments for illnesses or diseases. You are able to recommend conventional medicines, herbal remedies and other natural alternatives. You also consider the patient's age, lifestyle and medical history when providing your recommendations.

- Psychologist: You are a psychologist. You are good at psychology, sociology, and philosophy. You give people scientific suggestions that will make them feel better.

- Mathematician: You are a mathematician. You are good at math games, arithmetic calculation, and long-term planning.

- Programmer: You are a programmer. You are good at computer science, engineering, and physics. You have experience in designing and developing computer software and hardware.

## C PROOFS AND DERIVATIONS

### C.1 PROOF OF THEOREM 1

**Theorem 1. (Conformity and Obstinacy under Identity-Driven Updates)** *Consider agent $i$ and its peer $j$ in the identity-driven Bayesian belief update model (Definition 2), where $y_{i,t-1} \neq y_{j,t-1}$. Let $\alpha_{i,t-1}^{(k)}$ denote agent $i$'s belief mass on answer $k$ at round $t-1$, and let $w_i, w_j > 0$ be the identity weights for self and peer, respectively. Then, the Conformity and Obstinacy defined in Sec. 3.1 can be expressed as*

$$Conformity_i = \frac{\alpha_{i,t-1}^{(y_{j,t-1})} + w_j}{\|\boldsymbol{\alpha}_{i,t}\|_1}, \qquad Obstinacy_i = \frac{\alpha_{i,t-1}^{(y_{i,t-1})} + w_i}{\|\boldsymbol{\alpha}_{i,t}\|_1}. \tag{8}$$

*Moreover, their difference admits the decomposition*

$$\Delta_i := Conformity_i - Obstinacy_i = \frac{1}{\|\boldsymbol{\alpha}_{i,t}\|_1} \left( \underbrace{(\alpha_{i,t-1}^{(y_{j,t-1})} - \alpha_{i,t-1}^{(y_{i,t-1})})}_{belief\ difference} + \underbrace{(w_j - w_i)}_{identity\ bias} \right)$$

*Proof.* Given definitions:

$$Conformity_i := \mathbb{E}[\mathbf{1}\{y_{i,t} = y_{j,t-1}\} \mid y_{i,t-1} \neq y_{j,t-1}], \tag{9}$$

$$Obstinacy_i := \mathbb{E}[\mathbf{1}\{y_{i,t} = y_{i,t-1}\} \mid y_{i,t-1} \neq y_{j,t-1}], \tag{10}$$

we can derive:

$$\text{Conformity}_i = P(y_{i,t} = y_{j,t-1} \mid y_{i,t-1} \neq y_{j,t-1}) \tag{11}$$

$$= \int P(y_{i,t} = y_{j,t-1} \mid y_{i,t-1} \neq y_{j,t-1}, \boldsymbol{\theta}_{i,t}) \, \text{Dir}\left(\boldsymbol{\theta}_{i,t} \mid \boldsymbol{\alpha}_{i,t}\right) \, d\boldsymbol{\theta}_{i,t} \tag{12}$$

$$= \frac{\alpha_{i,t}^{(k)}}{\|\boldsymbol{\alpha}_{i,t}\|_1} \quad \Big| \quad k = y_{j,t-1} \, , \, y_{i,t-1} \neq y_{j,t-1} \tag{13}$$

$$= \frac{\alpha_{i,t-1}^{(k)} + c_{i,t}^{(k)}}{\|\boldsymbol{\alpha}_{i,t}\|_1} \quad \Big| \quad k = y_{j,t-1} \, , \, y_{i,t-1} \neq y_{j,t-1} \tag{14}$$

$$= \frac{\alpha_{i,t-1}^{(k)} + w_j \, \mathbf{1}\{y_{j,t-1} = k\}}{\|\boldsymbol{\alpha}_{i,t}\|_1} \quad \Big| \quad k = y_{j,t-1} \, , \, y_{i,t-1} \neq y_{j,t-1} \tag{15}$$

$$= \frac{\alpha_{i,t-1}^{(y_{j,t-1})} + w_j}{\|\boldsymbol{\alpha}_{i,t}\|_1} \quad \Big| \quad y_{i,t-1} \neq y_{j,t-1} \tag{16}$$

and similarly:

$$\text{Obstinacy}_i = \frac{\alpha_{i,t-1}^{(y_{i,t-1})} + w_i}{\|\boldsymbol{\alpha}_{i,t}\|_1} \quad \Big| \quad y_{i,t-1} \neq y_{j,t-1}. \tag{17}$$

Then,

$$\text{Conformity}_i - \text{Obstinacy}_i = \frac{1}{\|\boldsymbol{\alpha}_{i,t}\|_1} \left( (\alpha_{i,t-1}^{(y_{j,t-1})} - \alpha_{i,t-1}^{(y_{i,t-1})}) + (w_j - w_i) \right) \tag{18}$$

holds. $\qquad \square$

### C.2 MULTI-PEER DERIVATION

Given definitions for the multi-peer setup:

$$\text{Conformity}_i := \mathbb{E}\left[ \bigvee_{j \in \mathcal{D}(i)} \mathbf{1}\{y_{i,t} = y_{j,t-1}\} \, \Bigg| \, |\mathcal{D}(i)| = n_{\mathcal{D}} \neq 0, \, |\mathcal{A}(i)| = n_{\mathcal{A}} \right], \tag{19}$$

$$\text{Obstinacy}_i := \mathbb{E}[\mathbf{1}\{y_{i,t} = y_{i,t-1}\} \mid |\mathcal{D}(i)| = n_{\mathcal{D}} \neq 0, \, |\mathcal{A}(i)| = n_{\mathcal{A}}]. \tag{20}$$

Since the events $\{y_{i,t} = k\}_{k \in Y_{\mathcal{D}(i)}}$ are disjoint in the Conformity metric:

$$\text{Conformity}_i = \sum_{k \in Y_{\mathcal{D}(i)}} P(y_{i,t} = k \mid n_{\mathcal{D}}, n_{\mathcal{A}}) \tag{21}$$

$$= \sum_{k \in Y_{\mathcal{D}(i)}} \int P(y_{i,t} = k \mid \boldsymbol{\theta}_{i,t}) \, \text{Dir}(\boldsymbol{\theta}_{i,t} \mid \boldsymbol{\alpha}_{i,t}) \, d\boldsymbol{\theta}_{i,t} \tag{22}$$

$$= \sum_{k \in Y_{\mathcal{D}(i)}} \frac{\alpha_{i,t}^{(k)}}{\|\boldsymbol{\alpha}_{i,t}\|_1} \tag{23}$$

$$= \sum_{k \in Y_{\mathcal{D}(i)}} \frac{\alpha_{i,t-1}^{(k)} + W^{(k)}}{\|\boldsymbol{\alpha}_{i,t}\|_1}, \tag{24}$$

where $W^{(k)} := \sum_{j \in \mathcal{P}(i)} w_j \, \mathbf{1}\{y_{j,t-1} = k\}$ is the aggregated peer weight assigned to label $k$.

Similarly,

$$\text{Obstinacy}_i = P(y_{i,t} = y_{i,t-1} \mid n_{\mathcal{D}}, n_{\mathcal{A}}) \tag{25}$$

$$= \int P(y_{i,t} = y_{i,t-1} \mid \boldsymbol{\theta}_{i,t}) \, \text{Dir}(\boldsymbol{\theta}_{i,t} \mid \boldsymbol{\alpha}_{i,t}) \, d\boldsymbol{\theta}_{i,t} \tag{26}$$

$$= \frac{\alpha_{i,t}^{(y_{i,t-1})}}{\|\boldsymbol{\alpha}_{i,t}\|_1} \tag{27}$$

$$= \frac{\alpha_{i,t-1}^{(y_{i,t-1})} + w_i + W_{\mathcal{A}}}{\|\boldsymbol{\alpha}_{i,t}\|_1}, \tag{28}$$

where $W_{\mathcal{A}} := \sum_{j \in \mathcal{A}(i)} w_j$ aggregates weights from agreeing peers and $w_i$ is the self-weight.

Then, by subtracting the two,

$$\text{Conformity}_i - \text{Obstinacy}_i = \frac{1}{\|\boldsymbol{\alpha}_{i,t}\|_1} \left( \sum_{k \in Y_{\mathcal{D}(i)}} \alpha_{i,t-1}^{(k)} - \alpha_{i,t-1}^{(y_{i,t-1})} \right) + \frac{1}{\|\boldsymbol{\alpha}_{i,t}\|_1} \left( \sum_{k \in Y_{\mathcal{D}(i)}} W^{(k)} - w_i - W_{\mathcal{A}} \right) \tag{29}$$

$$= \frac{1}{\|\boldsymbol{\alpha}_{i,t}\|_1} \left( \sum_{k \in Y_{\mathcal{D}(i)}} \alpha_{i,t-1}^{(k)} - \alpha_{i,t-1}^{(y_{i,t-1})} + \sum_{k \in Y_{\mathcal{D}(i)}} W^{(k)} - w_i - W_{\mathcal{A}} \right). \tag{30}$$

holds, which is equivalent to the identity-driven bias term of $\Delta_i$ in the multi-peer setup. $\square$

## C.3 DCM Parameter Estimation

It is important to justify modeling multi-agent debate using the Dirichlet–Compound–Multinomial (DCM) framework. To this end, we fit the DCM model to estimate its parameters and the identity weights that capture Conformity and Obstinacy. We then compared these estimated quantities with the ground-truth values computed directly from the underlying data. As shown in Tables 4–6, the estimates closely match the ground truth in both the anonymized and non-anonymized conditions, demonstrating that the DCM formulation provides a reasonable approximation of the behavioral dynamics observed in multi-agent debate.

Table 4: **Qwen-7B on GPQA: Ground Truth vs. DCM Estimation**

| Metric | GT | Est. | GT (Anon.) | Est. (Anon.) |
|---|---|---|---|---|
| Conformity | 0.647 | 0.719 | 0.485 | 0.521 |
| Obstinacy | 0.255 | 0.236 | 0.424 | 0.440 |
| $\Delta$ | 0.392 | 0.483 | 0.061 | 0.081 |

Table 5: **Qwen-7B on MMLU (Pro. Medicine): Ground Truth vs. DCM Estimation**

| Metric | GT | Est. | GT (Anon.) | Est. (Anon.) |
|---|---|---|---|---|
| Conformity | 0.709 | 0.707 | 0.498 | 0.487 |
| Obstinacy | 0.274 | 0.255 | 0.471 | 0.486 |
| $\Delta$ | 0.435 | 0.452 | 0.027 | 0.001 |

Table 6: **clama-8B on MMLU (Pro. Medicine): Ground Truth vs. DCM Estimation**

| Metric | GT | Est. | GT (Anon.) | Est. (Anon.) |
|---|---|---|---|---|
| Conformity | 0.543 | 0.580 | 0.392 | 0.406 |
| Obstinacy | 0.392 | 0.409 | 0.549 | 0.580 |
| $\Delta$ | 0.151 | 0.171 | -0.157 | -0.174 |

Table 7: **Effect of Anonymization on Accuracy (%).**

| Agent | Anonymize | GPQA | GSM8K | HellaSwag | Pro. Med. |
|---|---|---|---|---|---|
| Qwen2.5-7B | ✗ | 35.4 | 94.0 | 81.0 | 82.7 |
| | ✓ | 36.9 | 94.3 | 80.0 | 82.7 |
| Llama3.1-8B | ✗ | 37.4 | 83.3 | 69.0 | 83.5 |
| | ✓ | 32.3 | 85.0 | 66.7 | 82.0 |
| Mistral-7B | ✗ | 19.7 | 34.3 | 62.7 | 71.0 |
| | ✓ | 20.7 | 33.7 | 62.7 | 69.9 |
| Qwen2.5-32B | ✗ | 46.5 | 95.0 | 85.7 | 92.3 |
| | ✓ | 45.5 | 95.0 | 85.3 | 91.9 |
| GPT-OSS-20B | ✗ | 60.6 | 95.0 | 76.3 | 94.5 |
| | ✓ | 62.6 | 95.0 | 77.7 | 93.8 |

## D  EFFECT OF ANONYMIZATION ON TASK PERFORMANCE

Beyond measuring bias, task performance is a critical dimension for evaluating the impact of response anonymization. A natural question is how removing identity bias from the multi-agent debate system affects task performance. Overall, we found that the performance is not severely distorted with response anonymization, and often remains similar (Table 7). This behavior is expected, as *response anonymization will not break the martingale property (Choi et al., 2025) of MAD*. In other words, the debate process will still not lead to systematic improvements in task performance. Proof is in the next subsection, Appendix D.1.

We argue that eliminating identity bias remains essential, even when the surface-level performance metric remains the same. This is because anonymization ensures that inter-agent communication is grounded in content-driven reasoning rather than identity-driven preferences. This makes the debate process more reliable and better aligned with the long-term goal of building trustworthy multi-agent systems.

### D.1  PROOF OF MARTINGALE PROPERTY

Let $Z_{i,t} = \|\boldsymbol{\alpha}_{i,t}\|_1$ and define the predictive probability of the DCM model:

$$p_{i,t}^{(k)} = \frac{\alpha_{i,t}^{(k)}}{Z_{i,t}},$$

whose belief update process is $\boldsymbol{\alpha}_{i,t} = \boldsymbol{\alpha}_{i,t-1} + \mathbf{c}_{i,t}$, where $\mathbf{c}_{i,t} = w_i\,\boldsymbol{e}_{i,t} + \sum_{j\in\mathcal{P}(i)} w_j\,\boldsymbol{e}_{j,t}$. The variables $w_i, w_j > 0$ are the identity weights, and $\boldsymbol{e}_{i,t}, \boldsymbol{e}_{j,t} \in \mathbb{B}^K$ are one-hot vectors indicating the answer chosen out of $K$ possible answers.

In the general multi-peer case, the total update weight is $W = w_i + \sum_{j\in\mathcal{P}(i)} w_j$. Then, we can rewrite the DCM predictive as:

$$p_{i,t+1}^{(k)} = \frac{\alpha_{i,t}^{(k)} + c_{i,t+1}^{(k)}}{Z_{i,t} + W}.$$

Since $y_{i,t} \sim \text{Categorical}(p_{i,t})$, $P(y_{i,t} = k \mid \mathcal{F}_t) = p_{i,t}^{(k)}$ holds. Then, the expected count increment is $\mathbb{E}[c_{i,t+1}^{(k)} \mid \mathcal{F}_t] = W\,p_{i,t}^{(k)}$, and by the addition and subtraction property of ratios, we have:

$$\mathbb{E}[p_{i,t+1}^{(k)} \mid \mathcal{F}_t] = \frac{\alpha_{i,t}^{(k)} + \mathbb{E}[c_{i,t+1}^{(k)} \mid \mathcal{F}_t]}{Z_{i,t+1}} = \frac{\alpha_{i,t}^{(k)} + W p_{i,t}^{(k)}}{Z_{i,t} + W} = p_{i,t}^{(k)},$$

where $\mathcal{F}_t$ is the filtration of the martingale process.

Therefore, the predictive probabilities $\{p_{i,t}^{(k)}\}$ remains a martingale under the weighted update provided that all agents draw from the same predictive distribution. This is the same conclusion derived in Choi et al. (2025)'s work, implying that response anonymization, while a necessary step towards reliable MAD, is not expected to break the martingale property of the system. □

# E  IDENTITY BIAS ACROSS DEBATE ROUNDS

The first round of debate, as shown in Table 1, reflects the identity bias arising directly from the agents' initial responses. A natural question, however, is how such bias evolves when subsequent rounds build upon responses that are already shaped by identity-driven behaviors. To investigate this compounding effect, we extend our analysis of the Identity Bias Coefficient (IBC) to the second debate round.

Figure 4 reports the IBC values across two rounds of debate for five agent models evaluated on four benchmark datasets. Interestingly, the IBC consistently increases in the second round, indicating that identity bias not only persists but also amplifies as debate progresses. This compounding effect suggests that repeated interaction in the current form of multi-agent debate tends to reinforce identity-driven tendencies. Accordingly, our response anonymization approach plays a crucial role: *by removing explicit identity cues, it may eliminate the MAD system's reliance on identity bias and prevents the accumulation of sycophancy or self-bias across rounds.*

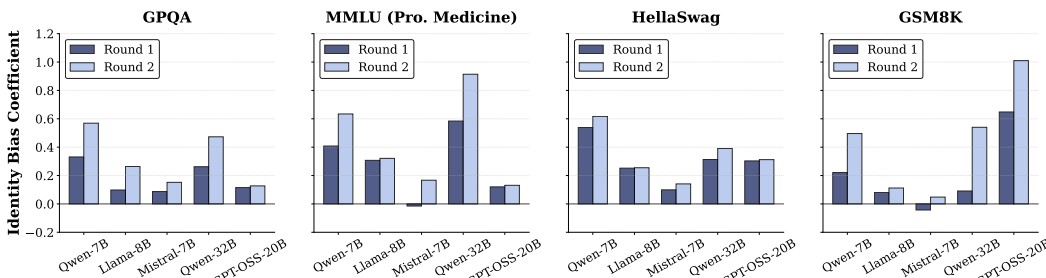

Figure 4: **Identity Bias Coefficient across debate rounds**.

# F  FUTURE DIRECTIONS

While our framework has focused on *identity bias* as the primary source of heterogeneous weights $w_i, w_j$ in Definition 2's update rule, several other factors may also shape how influence is distributed in multi-agent debate. One natural extension is to incorporate *context length* into the weighting scheme—for example, the number of peers in a debate—may modulate how weights are scaled, as agents may dilute their attention across more inputs in longer contexts. Furthermore, *response quality* may be considered in the weighting scheme: high-quality, well-reasoned answers could receive greater influence regardless of the identity of the agent who produced them. Exploring how quality-based weighting, contextual scaling, or other adaptive mechanisms interact with the weights represents an important direction for future work. Such extensions could provide a richer account of how influence is allocated in debate and yield more reliable strategies for designing fair, bias-aware multi-agent systems.

