# OpenReview forum: "Measuring and Mitigating Identity Bias in Multi-Agent Debate via Anonymization"
_ICLR.cc/2026/Conference — ICLR 2026 Conference Withdrawn Submission_

### Official Review · Reviewer_RJjL · 2025-10-28

**Soundness:** 1
**Presentation:** 4
**Contribution:** 2
**Rating:** 2
**Confidence:** 4

**Summary:**

The paper studies identity-driven biases in multi‑agent debate (MAD) with LLMs. Identity-driven biases are broken down to sycophancy (conformity, over-weighting peer opinions) and self-bias (obstanicy, over-weighting self opiniongs) which may distort collective reasoning. The authors formalize debate as a Bayesian update with Dirichlet-multinomial distribution, under which the gap = conformity - obstinacy  is a sum of a content‑driven belief difference term and a pure identity bias term. Then, a ‘response anonymization’ which removes all self / peer markers from the input prompts of next-round multi-agent debate will remove the identity bias term, allowing one to measure the term (named IBC). Experiments across different benchmarks and LLMs show mixed results, with tendency of language models to have positive IBC, indicating conformity typically has a larger effect than obstinacy.

**Strengths:**

1. Reproducibility and presentation quality. The paper provides detailed experiment settings including input prompts. Figures and tables are clean. It is a well-written and well-presented paper.

2. The idea of formulating LLM agent collective reasoning as a Bayesian update of Dirichlet prior is intriguing, and a finding of disproportionality in opinion update is interesting. This idea and finding is worth further investigation.

**Weaknesses:**

1. Statistical inference (Contribution 1, Line 76) is disconnected from the experiments and analysis. The paper’s core contribution is a Bayesian model of opinion updating in MAD. Yet the model yields no substantive insight beyond the self-explanatory intuition ‘masking identity in input prompts removes identity bias’. If the paper adopts a Bayesian framing (or more generally, statistical inference), it should lead to inference about the data-generating process: estimate parameters (e.g., \(\alpha, w_i, w_j\)) from data and assess the realism of the model. Currently the formalism serves mainly to restate an obvious implication—if \(w_i = w_j\) by identity masking in input prompts, then \(w_j - w_i = 0\) (Equation 4, Line 258) without empirically validating that the statistical model explains observed behavior.

A practical path would be to turn the formalism into an estimable model. One rough sketch might be like: under single-agent generation (Line 203) one could use empirical Bayes method to infer \(\alpha_{i,t}\) (or, the normalized masses \(\alpha_{i,t} / \sum \alpha_{i,t}\)) and estimate the identity-bias coefficient (Line 296) from result in Table 1. The paper could then test whether these inferred quantities jointly account for the observed conformity–obstinacy differences (Equation 4, Line 258). As it stands, one can skip Section 4 without losing interpretability of Tables 1-2 once conformity / obstinacy are defined (Sec. 3.1), which underscores the current disconnect. Demonstrating that the model fits and explains the generated data would substantially strengthen the contribution and value of the paper.

2. There seems to be a logical gap in the interpretation of experiment results with the formalism. When measuring the conformity - obstinacy gap after identity masking, the paper claims “As established in Corollary 1, once identity bias is removed, the residual Δ reflects only the difference in the agent’s prior belief masses on the peer’s versus its own previous answer. Thus, small but nonzero values after anonymization are expected” (Line 341-344) indicating this result can be explained by the model, which is not justified. In fact, the expectation of ‘Δ in the absence of identity bias’ (Corollary 1, Line 275) over a joint distribution of y_{i,t-1} and y_{j,t-1} should be zero under the identical agent setting (i.e. same persona for i and j) as in Table 1. This part requires clarification.

3. Identity masking hurts performance (Line 345; Table 3, Line 972). While the paper defends “eliminating identity bias remains essential” (Line 995) because it “makes the debate process more reliable and better aligned with the long-term goal of building trustworthy multi-agent” (Line 997), (1) the claim of reliability and trustworthiness is questionable and not demonstrated (2) some degree of identity-related behavior can be instrumental rather than purely harmful: for example, mild in-group / confirmation tendencies (e.g., confirmation bias) and socially productive forms of influence (e.g. accommodation in Communication Accommodation Theory, perspective-taking) can facilitate coordination and collective performance in human groups. Absent evidence that suppressing identity cues is essential, the paper’s blanket prescription to remove identity appears premature.

**Questions:**

Dirichlet distribution as a conjugate prior for categorical distribution is well applied to discrete probability measure; however, how can this be extended to a countably infinite outcome space, for example, integer answer space? While several benchmarks have a finite answer space (like MMLU 4-option questions), others (e.g. AIME accepts integer answer 0-999) have much larger space, and since LLM is capable of open-ended generation task, the outcome space can be countably infinite. I would like to ask how the current framework can be extended in this case.

Related to Weakness 1, I would like to ask how the prior and weights, which are hyperparameters of the paper’s statistical model, can be chosen. Definition 1 treats α as free, but expectations of y_{i,t} (marginalized over thetas) are scale‑invariant in α while Δ and IBC depend on the sum of α (denominator in Theorem 1). How do you infer prior and weights in experiments?

Recent work (e.g., Qian et al., To Mask or to Mirror: Human–AI Alignment in Collective Reasoning) examines identity bias in LLM multi-agent collective reasoning. I’m trying to understand the current state of this discussion: how well documented is identity-driven behavior in multi-agent setting, and to what extent is it a new observation versus an extension of earlier findings (e.g., LLMs favoring its own model generations) ?

---

> ### Author Response · Authors · 2025-11-20
>
> We thank the reviewer for the constructive feedback and for complimenting the quality of our work.
>
> Below, we address the key concerns.
>
> ---
>
>
> `A1` *Justifying the Dirichlet-Compound-Multinomial Model*
>
> Thank you for the great suggestions! Using the generated responses, we fitted the DCM model to estimate its parameters and identity weights that characterize Conformity and Obstinacy. We then compared these Conformity and Obstinacy estimates with the Ground Truth values derived directly from the underlying data (tables below).
>
> Overall, the estimated values align closely with the Ground Truth in both anonymized and non-anonymized settings, indicating that modeling multi-agent debate as a DCM process provides a reasonable approximation of the underlying behavioral dynamics. We have included this analysis in our updated manuscript.
>
> | **Qwen7B-GPQA** | Ground Truth | Estimation | Ground Truth (Anonymized) | Estimation (Anonymized) |
> |:---------------:|:-----------:|:----------:|:------------------------:|:-----------------------:|
> |    Conformity   |    0.647    |    0.719   |           0.485          |          0.521          |
> |    Obstinacy    |    0.255    |    0.236   |           0.424          |          0.440          |
> |      $\Delta$     |    0.392    |    0.483   |           0.061          |          0.081          |
>
> | **Qwen7B-MMLU (Pro. Med.)** | Ground Truth | Estimation | Ground Truth (Anonymized) | Estimation (Anonymized) |
> |:---------------------------:|:-----------:|:----------:|:------------------------:|:-----------------------:|
> |          Conformity         |    0.709    |    0.707   |           0.498          |          0.487          |
> |          Obstinacy          |    0.274    |    0.255   |           0.471          |          0.486          |
> |            $\Delta$          |    0.435    |    0.452   |           0.027          |          0.001          |
>
> | **Llama8B-MMLU (Pro. Med.)** | Ground Truth | Estimation | Ground Truth (Anonymized) | Estimation (Anonymized) |
> |:----------------------------:|:-----------:|:----------:|:------------------------:|:-----------------------:|
> |          Conformity          |    0.543    |    0.580   |           0.392          |          0.406          |
> |           Obstinacy          |    0.392    |    0.409   |           0.549          |          0.580          |
> |            $\Delta$            |    0.151    |    0.171   |          -0.157          |          -0.174         |
>
> ---
>
> `A2` *Clarifying the expected value of $\Delta$*
>
> After careful re-analysis, we confirm that the expected value of
>
> \begin{equation}
>     \Delta_i = \frac{\alpha_{i,t-1}^{(y_{j,t-1})} - \alpha_{i,t-1}^{(y_{i,t-1})} }{||{\alpha}\_{i,t}||_1}
> \end{equation}
>
> is indeed zero. In an MAD setting with two homogeneous agents, the relationship $\Delta_1 = -\Delta_2$ holds, which implies that the expectation of $\Delta_i$ is zero. We sincerely thank the reviewer for the careful and rigorous evaluation of our claims, and we have revised the manuscript accordingly. We apologize for any confusion this may have caused.
>
> Importantly, we assure that **this correction does not affect the main conclusions of our paper**. First, the empirically observed $\Delta$ in the anonymized condition (Table 1) remain close to zero, with an overall mean of -0.016 and a standard deviation of 0.076. Second, the Identity Bias Coefficient (IBC) continues to be well-motivated, as it analytically removes residual effects arising from belief differences, providing a more accurate measure of identity bias.

---

> ### Author Response · Authors · 2025-11-20
>
> ---
>
> `A3` *Demonstration of improvement in trustworthiness*
>
> We assess trustworthiness by analyzing two behavioral ratios--Subversion and Correction--defined as:
>
> - $\text{Subversion} = P\left[y_{i,t} = \text{incorrect}  \middle|  y_{i,t-1} = \text{correct},  y_{j,t-1} = \text{incorrect} \right].$
> - $\text{Correction} = P\left[ y_{i,t} = \text{correct}  \middle|  y_{i,t-1} = \text{incorrect},  y_{j,t-1} = \text{correct} \right].$
>
> By comparing these ratios before and after anonymization (tables below), we observe that **the Subversion ratio consistently exhibits a larger relative drop than the Correction ratio**. For instance, on MMLU (Professional Medicine) with Qwen-32B, the Subversion ratio decreases by 64.3%, whereas the Correction ratio decreases by only 14.9% after anonymization. This indicates that LLM agents are more prone to subverting their originally correct answers when identities are visible, and that Identity Anonymization effectively reduces such undesirable behavior.
>
> However, despite the larger proportional drop in Subversion, **its overall effect on total accuracy is mitigated by the much larger number of Correction events.** In other words, even though Subversion becomes significantly less frequent in ratio, the net accuracy impact is dominated by the greater volume of Correction cases, partially counteracting the benefit. We have included the full list of tables and analysis in the updated manuscript.
>
> | GPT-OSS-20B | Subversion Change | Subversion Drop Rate | Correction Change | Correction Drop Rate |
> |---|:---:|:---:|:---:|:---:|
> | GPQA | 0.164 $\rightarrow$ 0.091 | 44.5 \% | 0.882 $\rightarrow$ 0.864 | 2.0 \% |
> | GSM8K | 0.397 $\rightarrow$ 0.288 | 27.5 \% | 0.809 $\rightarrow$ 0.753 | 6.9 \% |
> | HellaSwag | 0.630 $\rightarrow$ 0.500 | 19.4 \% | 0.581 $\rightarrow$ 0.487 | 16.2 \% |
> | MMLU (Pro. Medicine) | 0.750 $\rightarrow$ 0.268 | 7.8 \% | 0.965 $\rightarrow$ 0.951 | 1.5 \% |
>
> | Qwen-32B | Subversion Change | Subversion Drop Rate | Correction Change | Correction Drop Rate |
> |---|:---:|:---:|:---:|:---:|
> | GPQA | 0.473 $\rightarrow$ 0.357 | 24.5 \% | 0.736 $\rightarrow$ 0.651 | 11.5 \% |
> | GSM8K | 0.455 $\rightarrow$ 0.333 | 26.8 \% | 0.727 $\rightarrow$ 0.727 | 0.0 \% |
> | HellaSwag | 0.630 $\rightarrow$ 0.500 | 20.6 \% | 0.739 $\rightarrow$ 0.543 | 26.5 \% |
> | MMLU (Pro. Medicine) | 0.750 $\rightarrow$ 0.268 | 64.3 \% | 0.839 $\rightarrow$ 0.714 | 14.9 \% |
>
>
> | Qwen-7B | Subversion Change | Subversion Drop Rate | Correction Change | Correction Drop Rate |
> |---|:---:|:---:|:---:|:---:|
> | GPQA | 0.717 $\rightarrow$ 0.500 | 30.3 \% | 0.711 $\rightarrow$ 0.553 | 22.2 \% |
> | GSM8K | 0.342 $\rightarrow$ 0.233 | 31.9 \% | 0.740 $\rightarrow$ 0.575 | 22.3 \% |
> | HellaSwag | 0.709 $\rightarrow$ 0.430 | 39.4 \% | 0.767 $\rightarrow$ 0.488 | 36.4 \% |
> | MMLU (Pro. Medicine) | 0.579 $\rightarrow$ 0.389 | 32.8 \% | 0.853 $\rightarrow$ 0.632 | 25.9 \% |

---

> ### Author Response · Authors · 2025-11-20
>
> ---
>
> `A4` *Can identity-related behavior sometimes be instrumental?*
>
> To evaluate whether agent identity can be useful in certain cases, we explicitly measured how often agents conform to a designated expert persona before and after anonymization. Concretely, on the MMLU Professional Medicine benchmark with heterogeneous Qwen-7B agents, we compared conformity toward a Doctor persona (intended "expert") and a generic Assistant persona in both the vanilla and anonymized MAD settings:
>
>
> | Peer Persona | Conform Rate (Non-Anonymized) | Conform Rate (Anonymized) | Drop Rate |
> |---|---|---|---|
> | Doctor (Expert) | 0.5217 | 0.3478 | 33.3% |
> | Assistant | 0.6429 | 0.4286 | 33.3% |
>
> We observe two key patterns:
>
> - **Anonymization reduces conformity at a similar rate for both personas**. The conform rate drops by 33.3% for both the Doctor and Assistant personas, suggesting that anonymization does not selectively suppress deference to the supposed expert; it uniformly dampens identity-driven conformity.
> - Even without anonymization, **agents do not preferentially defer to the "expert" persona**. In fact, the conform rate is higher toward the Assistant persona than toward the Doctor. A likely explanation is that the two personas had very similar single-agent performance (both around 80% accuracy on this benchmark), so the Doctor persona did not actually provide a clearly superior information source that would justify increased deference. When peer answers are comparably reliable, agents have little incentive to systematically favor one persona as an expert.
>
> > **But will identity-bias prove instrumental if there exists a true expert among agents?**
>
> Although exposing agent identities could occasionally accelerate consensus--such as when a true "expert" agent is present--we caution against relying on this effect. Any potential benefit is outweighed by the risk of identity-driven behavioral biases, including undue deference, subversion, or overconfidence tied to perceived status rather than evidence. In other words, even if identity cues sometimes provide a shortcut to agreement, they introduce undesirable and unpredictable dynamics that undermine robustness and trustworthiness. Thus, identity-based behavior is neither necessary nor sufficiently reliable to justify its risks. We insist that an agent's decision to defer or stand by its position should be grounded solely in the **content and quality of reasoning, without any reliance on identity information**.
>
> ---
>
> `A5` *Extension to the countably infinite answer space*
>
> We thank the reviewer for raising this important point. Our use of the DCM model with a finite answer space is intended as a tractable and interpretable abstraction of the belief update process. In practice, even when the theoretical outcome space is countably infinite (e.g., integer-valued tasks such as AIME), the effective response space--i.e., the set of answers actually produced by LLM agents--is finite and highly sparse. This makes a Dirichlet prior over the observed support a well-behaved and practically accurate approximation. Extending the framework to a genuinely countably infinite space is conceptually plausible: one may employ a Dirichlet process (DP) prior or other nonparametric Bayesian constructions, which generalize Dirichlet conjugacy to distribution-level infinite partitions. We will clarify this modeling choice and include a brief discussion of possible nonparametric extensions in the revised version.
>
>
> ---
>
> `A6` *Definition 1 treats $\alpha$ as free, but ... How do you infer prior and weights in experiments?*
>
> Thank you for this insightful question! To ensure more precise comparison across settings, we may calibrate the IBC values using the estimated magnitude of the Dirichlet concentration, retrieved with the same procedure described in `A1`:
> \begin{equation}
> \text{IBC}\_\text{calibrated} = \text{IBC} \times  ||\hat{\alpha}_{i,t}||_1 = \frac{w_j - w_i}{{||{\alpha}\_{i,t}||}\_1} \times {||\hat{\alpha}\_{i,t}||}\_1
> \end{equation}

---

> ### Author Response · Authors · 2025-11-20
>
> ---
>
> `A7` *Novelty*
>
> We appreciate the reviewer's question regarding the broader landscape of identity-driven behavior in LLMs. Prior work has documented related phenomena in **single-agent settings**, such as self-preferential evaluation or  sycophancy, but systematic analysis in **interactive multi-agent reasoning** remains sparse. The concurrent work by Qian et al. (released on arXiv after our ICLR submission) provides an important complementary perspective on identity cues, but focuses primarily on demographic and pseudonym effects in human–AI mixed groups rather than modeling agent-to-agent belief dynamics.
>
> In contrast, our work is, to our knowledge, the first to unify sycophantic behaviors and self-biased tendencies into a single spectrum based on a probabilistic framework, showing how identity cues shape multi-agent debate dynamics. Our DCM-based formulation, the Identity Bias Coefficient, and our systematic comparison of identity-visible vs. anonymized agents go beyond earlier findings by quantifying how identity alters the evolution of beliefs during collective reasoning.

---

### Official Review · Reviewer_eSo1 · 2025-10-31

**Soundness:** 3
**Presentation:** 3
**Contribution:** 3
**Rating:** 6
**Confidence:** 3

**Summary:**

This paper addresses the phenomenon of identity-bias in multi-agent debate (MAD) systems of large language models (LLMs). In such systems, multiple agents generate answers, see each other’s responses, revise, and then aggregate the result. The authors observe that agents tend to behave differently depending on whether they are looking at their own prior response (“self”) or a peer’s (“peer”) in particular exhibiting sycophancy (over-adopting peer responses) or self-bias (sticking too much to one’s prior answer). They propose a formal probabilistic Bayesian belief-update model that explicitly incorporates identity weighting to capture these biases. Based on that, they define interpretable metricsand a derived metric Identity Bias Coefficient (IBC) to measure how much identity influences agent behaviour. Their key intervention is “Response Anonymization” (i.e., removing identity markers so agents cannot tell whether a previous answer was from self or peer), thereby forcing symmetric weighting and reducing identity-bias. Empirical experiments across several model families and tasks show that identity bias (especially sycophancy) is widespread in MAD, and that anonymization substantially reduces the bias. The paper concludes that anonymization is a lightweight but effective method for making multi-agent debate more content-driven rather than identity-driven.

**Strengths:**

- The paper tackles a relatively under-explored issue in the emerging area of multi-agent collaborative LLM reasoning, namely how agent identity influences dynamics rather than purely content.

- The formalisation is clear and elegant: modelling the agent’s belief updating as a Dirichlet-Multinomial (DCM) process and then deriving expressions that separate belief-driven update from identity-driven weight differences.
The proposed metrics (Conformity, Obstinacy, IBC) provide interpretable ways to quantify how much an agent is influenced by peer vs self, which is useful for analyzing such systems.

- Response Anonymization is conceptually simple but practically appealing: it requires no retraining or architecture change, just modifying how prompts are constructed. That makes it widely applicable.

- The empirical evaluation is reasonably broad: multiple model families, multiple datasets, both homogeneous and heterogeneous agent settings, and both single-peer and multi-peer setups. The results clearly show the presence of sycophancy and substantial reduction in bias under anonymization.

**Weaknesses:**

- The evaluation focuses on many standard reasoning tasks (GPQA, MMLU, HellaSwag, GSM8K) but may not cover more open-ended or real-world debate scenarios (e.g., multi-turn dialogues, adversarial peers, diverse agent personas beyond simple roles). The generalisability to those contexts may be unclear.
- The paper shows reduction in identity bias via anonymization but less discussion (or empirical depth) about how anonymization interacts with overall performance (accuracy, quality of final answer). There may be trade-offs (e.g., anonymization might reduce beneficial peer influence) that are not deeply explored.
- The intervention (anonymization) treats all identity cues as equal, but there may be scenarios where knowing “this answer came from a more expert agent” is beneficial (i.e., the identity might encode expertise). The paper does not deeply explore heterogeneous expertise settings or when identity cues might be legitimately informative.
- The paper missed some of the relevant reference such as https://arxiv.org/abs/2506.12657 in understanding how identity changes stances of multi-agent debate.

**Questions:**

- How does anonymization affect the accuracy or quality of the final aggregated answer in multi-agent debate? If anonymization reduces identity bias but also reduces correct consensus or slows convergence, that would modify how strongly I view this as an unequivocal improvement.
- In heterogeneous-agent settings where some agents are known to be more expert (e.g., a “doctor” vs “student” agent), how does anonymization affect outcomes? Does removing identity cues reduce the ability of the system to leverage expert peers?
- How stable are the results across more challenging debate formats (longer chains, adversarial peers, mixed objectives) or with more than two rounds of debate? If identity bias re-emerges in more complex settings, that would temper the generality of the findings.

---

> ### Author Response · Authors · 2025-11-20
>
> We thank the reviewer for the constructive feedback and for complimenting the novelty, practicality, theoretical rigor, and the extensiveness of our analyses.
>
> Below, we address the key concerns.
>
> ---
>
> `A1` *Evaluation setting*
>
> We appreciate the reviewer’s observation. Our primary focus in this work is to establish a clear and controlled empirical foundation for evaluating debate methods. For this reason, we prioritized standard reasoning benchmarks such as GPQA, MMLU, HellaSwag, and GSM8K, where ground truth exists and comparisons are reliable. While we agree that open-ended, real-world debate settings are an important direction, _systematically evaluating these scenarios presents both methodological and measurement challenges_: (1) there is no widely accepted protocol for scoring correctness or argumentative quality in such settings, and (2) results can be highly sensitive to prompt design and evaluator choice.
>
> Given these limitations, we opted to first ensure that our approach is rigorously validated in settings where evaluation noise is minimal and replicability is high. We view this as a necessary first step toward the broader goal of robust open-ended debate. We have already begun exploring extensions to more free-form tasks, but these require careful experimental design. We will clarify this in the revised manuscript.
>
>
> ---
>
> `A2` *How Anonymization affects performance, and trade-offs*
>
> To examine how identity anonymization interacts with overall performance, we analyze its effect on two key behavioral ratios: Subversion and Correction, defined as
>
>
> - $\text{Subversion} = P\left[y_{i,t} = y_{j,t-1}  \middle|  y_{i,t-1} = \text{correct},  y_{j,t-1} = \text{incorrect} \right].$
> - $\text{Correction} = P\left[ y_{i,t} = y_{j,t-1}  \middle|  y_{i,t-1} = \text{incorrect},  y_{j,t-1} = \text{correct} \right].$
>
> These quantities directly determine when agents abandon correct answers (harmful influence) or recover from incorrect ones (beneficial influence).
>
>
> Across all models and benchmarks (tables below), we observe that **anonymization consistently leads to a larger proportional decrease in Subversion than in Correction**. For example, on MMLU (Professional Medicine) with Qwen-32B, Subversion drops by 64.3%, while Correction decreases by only 14.9%. This suggests that anonymization meaningfully reduces harmful peer influence, while largely preserving the beneficial corrective behaviors that support productive debate.
>
> At the same time, our results reveal an important nuance regarding accuracy trade-offs: although the ratio of Subversion decreases more sharply, the overall number of Correction events is much larger in typical debates. As a result, **modest reductions in Correction can partially offset the gains from reduced Subversion when measuring total accuracy**. In other words, anonymization mitigates identity-driven distortions of belief dynamics, but its net effect on final accuracy depends on the fact that most beneficial peer influence occurs through Correction events. We have expand this discussion in the revised manuscript to clarify when anonymization yields accuracy improvements and when the effects are more balanced.
>
> | GPT-OSS-20B | Subversion Change | Subversion Drop Rate | Correction Change | Correction Drop Rate |
> |---|:---:|:---:|:---:|:---:|
> | GPQA | 0.164 $\rightarrow$ 0.091 | 44.5 \% | 0.882 $\rightarrow$ 0.864 | 2.0 \% |
> | GSM8K | 0.397 $\rightarrow$ 0.288 | 27.5 \% | 0.809 $\rightarrow$ 0.753 | 6.9 \% |
> | HellaSwag | 0.630 $\rightarrow$ 0.500 | 19.4 \% | 0.581 $\rightarrow$ 0.487 | 16.2 \% |
> | MMLU (Pro. Medicine) | 0.750 $\rightarrow$ 0.268 | 7.8 \% | 0.965 $\rightarrow$ 0.951 | 1.5 \% |
>
> | Qwen-32B | Subversion Change | Subversion Drop Rate | Correction Change | Correction Drop Rate |
> |---|:---:|:---:|:---:|:---:|
> | GPQA | 0.473 $\rightarrow$ 0.357 | 24.5 \% | 0.736 $\rightarrow$ 0.651 | 11.5 \% |
> | GSM8K | 0.455 $\rightarrow$ 0.333 | 26.8 \% | 0.727 $\rightarrow$ 0.727 | 0.0 \% |
> | HellaSwag | 0.630 $\rightarrow$ 0.500 | 20.6 \% | 0.739 $\rightarrow$ 0.543 | 26.5 \% |
> | MMLU (Pro. Medicine) | 0.750 $\rightarrow$ 0.268 | 64.3 \% | 0.839 $\rightarrow$ 0.714 | 14.9 \% |
>
>
> | Qwen-7B | Subversion Change | Subversion Drop Rate | Correction Change | Correction Drop Rate |
> |---|:---:|:---:|:---:|:---:|
> | GPQA | 0.717 $\rightarrow$ 0.500 | 30.3 \% | 0.711 $\rightarrow$ 0.553 | 22.2 \% |
> | GSM8K | 0.342 $\rightarrow$ 0.233 | 31.9 \% | 0.740 $\rightarrow$ 0.575 | 22.3 \% |
> | HellaSwag | 0.709 $\rightarrow$ 0.430 | 39.4 \% | 0.767 $\rightarrow$ 0.488 | 36.4 \% |
> | MMLU (Pro. Medicine) | 0.579 $\rightarrow$ 0.389 | 32.8 \% | 0.853 $\rightarrow$ 0.632 | 25.9 \% |

---

> ### Author Response · Authors · 2025-11-20
>
> ---
>
> `A3` *Does anonymization reduce the ability to leverage expert peers?*
>
> Interesting question! We explicitly measured how often agents conform to a designated "expert" persona before and after anonymization. Concretely, on the MMLU Professional Medicine benchmark with heterogeneous Qwen-7B agents, we compared the conform rate toward two peer personas: a Doctor persona (an "expert") and a generic Assistant persona, under both the vanilla MAD setting and the anonymized MAD setting. The results are shown below:
>
> | Peer Persona | Conform Rate (Non-Anonymized) | Conform Rate (Anonymized) | Drop Rate |
> |---|---|---|---|
> | Doctor (Expert) | 0.5217 | 0.3478 | 33.3% |
> | Assistant | 0.6429 | 0.4286 | 33.3% |
>
> We observe two key patterns:
>
>
> 1. **Anonymization reduces conformity at a similar rate for both personas**. The conform rate drops by 33.3% for both the Doctor and Assistant personas, suggesting that anonymization does not selectively suppress deference to the supposed expert; it uniformly dampens identity-driven conformity.
> 2. Even without anonymization, **agents do not preferentially defer to the "expert" persona**. In fact, the conform rate is higher toward the Assistant persona than toward the Doctor. A likely explanation is that the two personas had very similar single-agent performance (both around 80% accuracy on this benchmark), so the Doctor persona did not actually provide a clearly superior information source that would justify increased deference. When peer answers are comparably reliable, agents have little incentive to systematically favor one persona as an expert.
>
>
> Taken together, these findings suggest that, in our current setup, **anonymization does not meaningfully impair the ability to leverage expert peers**, since the system shows no strong expert-preferring behavior even in the non-anonymized condition. It would, however, be valuable to examine scenarios in which agents differ more substantially in true expertise. We leave this for future work.
>
> ---
>
> `A4` *Can identity-driven behavior sometimes be beneficial?*
>
> Although exposing agent identities could occasionally accelerate consensus--such as when a true "expert" agent is present--we caution against relying on this effect. Any potential benefit can also be outweighed by the risk of identity-driven behavioral biases, including undue deference, subversion, or overconfidence tied to perceived status rather than evidence. In other words, even if identity cues sometimes provide a shortcut to agreement, they can introduce undesirable and unpredictable dynamics that undermine robustness and trustworthiness. Thus, identity-based behavior is neither necessary nor sufficiently reliable to justify its risks. We believe that **an agent's decision to defer or stand by its position should be grounded primarily in the content and quality of reasoning, rather than reliance on identity information.**
>
> ---
>
> `A5` *Inclusion of additional references*
>
> Thank you for raising this point! The mentioned work is indeed highly relevant, particularly for settings with heterogeneous agents assigned distinct personas. We have now incorporated this and other related studies into our Related Work section.
>
> ---
>
> `A6` *Extension to multiple debate rounds*
>
> Below, we extend the analysis to three debate rounds and report the resulting $\Delta$ values along with their differences (i.e., IBC). Across all settings, we observe a substantial reduction in identity bias under anonymization. This demonstrates that our approach generalizes robustly to more complex scenarios, including longer debate sequences.
>
> | GPQA | **Round 1** | **Round 2** | **Round 3** |
> |---|:---:|:---:|:---:|
> | Llama-8B | 0.124 | 0.162 | -0.021 |
> | Llama-8B w/ Anonymization | 0.026 | -0.102 | -0.028 |
> | $\qquad$**IBC** | 0.098 | 0.264 | 0.007 |
> | Mistral-7B | 0.005 | -0.034 | 0.030 |
> | Mistral-7B w/ Anonymization | -0.082 | -0.186 | -0.190 |
> | $\qquad$**IBC** | 0.087 | 0.152 | 0.220 |
> | Qwen-7B | 0.392 | 0.472 | 0.421 |
> | Qwen-7B w/ Anonymization | 0.061 | -0.097 | -0.062 |
> | $\qquad$**IBC** | 0.331 | 0.569 | 0.483 |
> | Qwen-32B | 0.298 | 0.508 | 0.614 |
> | Qwen-32B w/ Anonymization | 0.036 | 0.035 | 0.009 |
> | $\qquad$**IBC** | 0.262 | 0.473 | 0.605 |
> | GPT-OSS-20B | 0.040 | 0.148 | 0.306 |
> | GPT-OSS-20B w/ Anonymization | -0.036 | -0.057 | -0.028 |
> | $\qquad$**IBC** | 0.076 | 0.205 | 0.334 |

---

### Official Review · Reviewer_u3fx · 2025-10-31

**Soundness:** 3
**Presentation:** 3
**Contribution:** 2
**Rating:** 4
**Confidence:** 3

**Summary:**

The authors systematically examine identity biases in multi-agent debates, specifically sycophancy (the tendency to adopt the opinions of other agents) and self-bias (the tendency to maintain one's own opinion). They formalize debate dynamics as a Bayesian update process, which explains the aforementioned tendencies, and propose a simple intervention to reduce identity biases. They also introduce the Identity Bias Coefficient (IBC) as a metric to measure identity biases.

**Strengths:**

The paper is well structured and easy to follow.
Response anonymization is a simple yet interesting and effective approach to reduce identity biases of LLMs in multi agent debates.

**Weaknesses:**

Response anonymization appears to have no effect, or rather a negative effect, on the overall model's accuracy.
The related work section suggests that other mitigation strategies have been investigated by previous studies, but does not specify which ones. This information is important for the paper.

**Questions:**

- What is the greatest benefit of removing identity biases in the case of multi agent debate?

---

> ### Author Response · Authors · 2025-11-20
>
> We thank the reviewer for the constructive feedback and for complimenting the practicality and effectiveness of our approach.
>
> Below, we address the key concerns.
>
> ---
>
>
> `A1` *Outline of previous mitigation strategies in the Related Works*
>
> Thank you for this thoughtful question. We would like to clarify that **the "mitigation strategies" discussed in our Related Work primarily target single-agent settings and do not model the interactive dynamics of multiple LLM agents**. To our knowledge, the only explicitly proposed mitigation method for identity-driven behavior in this space is a prompt-curation–based approach for reducing sycophancy [1]. While valuable, this work (i) requires LLM training, and (ii) its results cannot be easily reproduced without open-sourced code. In contrast, our method is (1) an inference-time method that is easy to implement, and (2) theoretically guaranteed to eliminate identity bias. Hence, we argue that our work is **the first unified, principled, and perfect mitigation mechanism for identity bias mitigation** rather than a heuristic reduction of bias.
>
>
> [1] P. Pitre et al. Consensagent: Towards efficient and effective consensus in multi-agent llm interactions through sycophancy mitigation. ACL 2025
>
> ---
>
> `A2-1` *What is the benefit of removing identity bias?*
>
> We appreciate the reviewer’s concern. Our aim is not to boost benchmark scores but to eliminate identity-driven distortions while maintaining comparable performance. More importantly, removing identity cues provides three key benefits for multi-agent debate:
>
> **Benefit 1: Enhance content-centric reasoning.**
> Identity cues cause agents to overweight either their own responses (self-bias) or their peers’ responses (sycophancy). Anonymization ensures that debate decisions are driven by argument content, not by who produced the argument. This makes the mechanism itself more reliable, regardless of its effect on accuracy.
>
> **Benefit 2: Improves trustworthiness and causal clarity of the debate mechanism.**
> Identity bias introduces a confound: a correct answer may emerge either from good reasoning or from preferential weighting of certain agents. By removing identity cues, we obtain cleaner, more causally meaningful belief-update trajectories. This makes it easier to analyze, debug, and trust multi-agent debate protocols—especially important for safety and mechanistic understanding.
>
> **Benefit 3: Removing identity largely preserves the accuracy**
> Our experiments (Table 3) show that accuracy remains comparable in most settings. Small ±1-2 point variations fall within normal model variance and reflect minor prompt perturbations. This demonstrates that identity-driven behavior can be removed at essentially minimum performance cost. In other words, anonymization is a low-risk mitigation that eliminates a systematic bias channel.
>
> In the following `A2-2`, we also provide further explanations on the intricate link between trustworthiness and accuracy.

---

> ### Author Response · Authors · 2025-11-20
>
> ---
>
> `A2-2` **Why improvement in Trustworthiness may not directly translate to higher accuracy?**
>
> We assess trustworthiness by analyzing two behavioral ratios--Subversion and Correction--defined as:
>
> - $\text{Subversion} = P\left[y_{i,t} = \text{incorrect}  \middle|  y_{i,t-1} = \text{correct},  y_{j,t-1} = \text{incorrect} \right].$
> - $\text{Correction} = P\left[ y_{i,t} = \text{correct}  \middle|  y_{i,t-1} = \text{incorrect},  y_{j,t-1} = \text{correct} \right].$
>
> By comparing these ratios before and after anonymization (tables below), we observe that **the Subversion ratio consistently exhibits a larger relative drop than the Correction ratio**. For instance, on MMLU (Professional Medicine) with Qwen-32B, the Subversion ratio decreases by 64.3%, whereas the Correction ratio decreases by only 14.9% after anonymization. This indicates that LLM agents are more prone to subverting their originally correct answers when identities are visible, and that Identity Anonymization effectively reduces such undesirable behavior.
>
>
> Although Subversion drops more sharply, **its overall effect on total accuracy can be mitigated by the much larger number of Correction events.** Thus, small reductions in Correction can partially offset the benefits of reduced Subversion. In short, anonymization makes the debate more principled and less identity-driven, but the net accuracy effect depends on the relative volume of corrective steps the agents naturally take during debate.
>
> | GPT-OSS-20B | Subversion Change | Subversion Drop Rate | Correction Change | Correction Drop Rate |
> |---|:---:|:---:|:---:|:---:|
> | GPQA | 0.164 $\rightarrow$ 0.091 | 44.5 \% | 0.882 $\rightarrow$ 0.864 | 2.0 \% |
> | GSM8K | 0.397 $\rightarrow$ 0.288 | 27.5 \% | 0.809 $\rightarrow$ 0.753 | 6.9 \% |
> | HellaSwag | 0.630 $\rightarrow$ 0.500 | 19.4 \% | 0.581 $\rightarrow$ 0.487 | 16.2 \% |
> | MMLU (Pro. Medicine) | 0.750 $\rightarrow$ 0.268 | 7.8 \% | 0.965 $\rightarrow$ 0.951 | 1.5 \% |
>
> | Qwen-32B | Subversion Change | Subversion Drop Rate | Correction Change | Correction Drop Rate |
> |---|:---:|:---:|:---:|:---:|
> | GPQA | 0.473 $\rightarrow$ 0.357 | 24.5 \% | 0.736 $\rightarrow$ 0.651 | 11.5 \% |
> | GSM8K | 0.455 $\rightarrow$ 0.333 | 26.8 \% | 0.727 $\rightarrow$ 0.727 | 0.0 \% |
> | HellaSwag | 0.630 $\rightarrow$ 0.500 | 20.6 \% | 0.739 $\rightarrow$ 0.543 | 26.5 \% |
> | MMLU (Pro. Medicine) | 0.750 $\rightarrow$ 0.268 | 64.3 \% | 0.839 $\rightarrow$ 0.714 | 14.9 \% |
>
>
> | Qwen-7B | Subversion Change | Subversion Drop Rate | Correction Change | Correction Drop Rate |
> |---|:---:|:---:|:---:|:---:|
> | GPQA | 0.717 $\rightarrow$ 0.500 | 30.3 \% | 0.711 $\rightarrow$ 0.553 | 22.2 \% |
> | GSM8K | 0.342 $\rightarrow$ 0.233 | 31.9 \% | 0.740 $\rightarrow$ 0.575 | 22.3 \% |
> | HellaSwag | 0.709 $\rightarrow$ 0.430 | 39.4 \% | 0.767 $\rightarrow$ 0.488 | 36.4 \% |
> | MMLU (Pro. Medicine) | 0.579 $\rightarrow$ 0.389 | 32.8 \% | 0.853 $\rightarrow$ 0.632 | 25.9 \% |

---

> > ### Comment · Reviewer_u3fx · 2025-11-24
> > **Answer to Official Comment by Authors**
> >
> > Dear Authors, thank you very much for addressing my concerns and providing additional analyses on trustworthiness. It is concerning that the correction rates seem to have dropped significantly in some cases. Nevertheless, I understand the importance of the approach to mitigate identity-bias in multi agent debates and will adjust my score accordingly.

---

> > > ### Author Response · Authors · 2025-11-24
> > >
> > > Dear Reviewer u3fx,
> > >
> > > Thank you for your response and for re-evaluating our work!
> > >
> > > Regarding the correction rate: the decrease itself is not problematic because (1) we interpret this behavior as a **rectifying effect of identity anonymization**---agents become less inclined to "correct" themselves in ways driven by identity-based biases, and (2) this decrease is **offset by a larger reduction in the Subversion rate**, leading to an overall improvement in trustworthiness.
> > >
> > > Other than this point, if there are any additional questions or clarifications that come to mind, please feel free to let us know.
> > > We thank you again for taking your time to review our work.
> > >
> > > Best regards,
> > >
> > > Authors

---

### Official Review · Reviewer_quLh · 2025-10-31

**Soundness:** 2
**Presentation:** 3
**Contribution:** 2
**Rating:** 4
**Confidence:** 3

**Summary:**

This paper addresses identity bias in multi-agent debate (MAD) systems, where LLM agents exhibit either sycophancy/conformity or self-bias/obstinacy. The authors formalize debate mechanics in LLMs as an identity-weighted Bayesian update process using a Dirichlet-Compound-Multinomial model. To mitigate the impact of identity bias on model responses, they propose response anonymization, which removes identity markers from prompts. The paper also introduces the Identity Bias Coefficient (IBC) to quantify the magnitude of identity bias. Comprehensive experiments demonstrate that identity bias is widespread, with sycophancy being more prevalent than self-bias, and that anonymization effectively reduces this bias.

**Strengths:**

The concept of unifying sycophancy and self-bias in MAD into identity bias is a novel proposal, which is supported by precise theoretical modeling using identity-weighted Bayesian updates in a principled and interpretable manner.

The response anonymization mitigation solution requires no retraining, architectural changes, or auxiliary losses and is shown to be consistent and effective across settings.

The authors demonstrate the robustness of their results across multiple models, tasks, and go beyond to include multi-peer and heterogeneous persona settings which makes the analysis compelling.

**Weaknesses:**

While the theoretical derivation is able to isolate an identity-bias term, the empirical demonstrations are not fully convincing as causal in nature. When anonymization reduces the Conformity-Obstinacy gap, is this necessarily because identity bias was eliminated? Could this be instead because anonymization changes the cognitive load or prompt complexity? The ordering of responses in anonymised prompts could also introduce noise which could separately impact model reasoning. Comparing anonymisation to other intervention methods such as randomly swapping labels, applying counterfactual identity relabeling with content being held constant would help to strengthen the claims made in the paper.

The paper defers how anonymization impacts task accuracy to the appendix. Table 3 even shows mixed performance effects of anonymization, but the discussion is quite brief. While the paper argues that eliminating bias is valuable regardless of performance, a deeper analysis of when and why anonymization hurts performance would be valuable to readers.

The belief difference term which represents reasoning driven by actual content differences warrants more analysis. While references are made to the empirical behaviour of this term in Section 5.2, the calibration and quality of this term is not fully examined. Without this, we also don’t know when anonymization reduces bias while deference to more accurate peers could actually be more beneficial.

In the heterogenous setting, the role of adding multiple specialised personas is under-explored as they may be relevant to task success. Tied to the previous point, in this case, anonymisation may lead to more substantial performance tradeoffs as domain-expert personas should likely attract conformity on domain specific items.

**Questions:**

Can you run alternate intervention experiments with counterfactual or random identity label swapping and compare the findings wrt point 1 mentioned above?

Could you report more analysis of when anonymization can lead to trade-offs in actual task performance? Are there predictable patterns based on task characteristics, model size, or debate configuration?

Could you empirically show how the belief difference term relates to actual content driven reasoning? Maybe a correlation analysis between proxies of model confidence (e.g. log probs) would make it more clear.

What is the impact of anonymisation in cases where knowing the identity is actually beneficial for the outcome, such as in domain-relevant personas are used (e.g. a doctor in the medical domain)? A justification in the context of debate dynamics could also be useful.

---

> ### Author Response · Authors · 2025-11-20
>
> We thank the reviewer for the constructive feedback and for complimenting the novelty, practicality, theoretical rigor, and the extensiveness of our analysis.
>
> Below, we address the key concerns.
>
> ---
>
> `A1` *Comparison with label-swapped MAD*
>
> Thank you for suggesting this interesting experiment! As suggested, we evaluated Conformity and Obstinacy in a setting where the identity labels of the "self" and "peer" agents are swapped (MAD-swapped). The table below compares the vanilla, swapped, and anonymized conditions on the MMLU Professional Medicine benchmark.
>
> Strikingly, when identities are swapped, the Conformity and Obstinacy values also swap, demonstrating that the debate dynamics are indeed identity-driven rather than artifacts of the task or underlying correctness patterns. In contrast, anonymizing identity information substantially closes the gap between Conformity and Obstinacy, confirming that identity cues are the primary source of this asymmetry.
>
> | Qwen2.5-7B | Conformity | Obstinacy | $\Delta$ |
> |---|:---:|:---:|:---:|
> | MAD-vanilla | 0.709 | 0.274 | 0.435 |
> | MAD-swapped | 0.224 | 0.753 | -0.529 |
> | MAD-anonymized | 0.498 | 0.471 | 0.027 |
>
> ---
>
> `A2` *Understanding the impact of anonymization on task accuracy*
>
>
> To examine when and how identity anonymization interacts with overall performance, we analyze its effect on two key behavioral ratios: Subversion and Correction, defined as
>
>
> - $\text{Subversion} = P\left[y_{i,t} = \text{incorrect}  \middle|  y_{i,t-1} = \text{correct},  y_{j,t-1} = \text{incorrect} \right].$
> - $\text{Correction} = P\left[ y_{i,t} = \text{correct}  \middle|  y_{i,t-1} = \text{incorrect},  y_{j,t-1} = \text{correct} \right].$
>
> These quantities directly determine when agents abandon correct answers (harmful influence) or recover from incorrect ones (beneficial influence).
>
> Across all models and benchmarks (tables below), anonymization produces a consistent and substantial reduction in Subversion, while its impact on Correction is much smaller. For example, on MMLU (Professional Medicine) with Qwen-32B, Subversion decreases by 64.3%, whereas Correction decreases by only 14.9%. This pattern generalizes across models of different sizes and across tasks with varying difficulty: anonymization reliably suppresses harmful identity-driven deference while largely preserving agents' ability to correct mistakes when peers provide better reasoning.
>
> However, our analysis also reveals predictable trade-offs. Even though the ratio of Subversion drops more sharply, if the absolute number of Correction events is typically much larger in multi-agent debate, even **modest reductions in Correction ratio can partially offset the gains from reduced Subversion**, yielding neutral or mixed effects on final accuracy. We expect this phenomenon will be emphasized in a task with a high density of correction opportunities (e.g., in tasks where agents initially disagree more often).
>
> We also observe **model-size-dependent patterns**:
> - Larger models (e.g., Qwen-32B, GPT-OSS-20B) maintain more stable Correction under anonymization, leading to clearer gains in trustworthiness and accuracy.
> - Smaller models (e.g., Qwen-7B) depend more heavily on peer influence, so anonymization has a smaller net effect on trustworthiness improvement.
>
> We have included these discussions in the manuscript to more clearly delineate when anonymization would improve performance and when trade-offs may arise.
>
> | GPT-OSS-20B | Subversion Change | Subversion Drop Rate | Correction Change | Correction Drop Rate |
> |---|:---:|:---:|:---:|:---:|
> | GPQA | 0.164 $\rightarrow$ 0.091 | 44.5 \% | 0.882 $\rightarrow$ 0.864 | 2.0 \% |
> | GSM8K | 0.397 $\rightarrow$ 0.288 | 27.5 \% | 0.809 $\rightarrow$ 0.753 | 6.9 \% |
> | HellaSwag | 0.630 $\rightarrow$ 0.500 | 19.4 \% | 0.581 $\rightarrow$ 0.487 | 16.2 \% |
> | MMLU (Pro. Medicine) | 0.750 $\rightarrow$ 0.268 | 7.8 \% | 0.965 $\rightarrow$ 0.951 | 1.5 \% |
>
> | Qwen-32B | Subversion Change | Subversion Drop Rate | Correction Change | Correction Drop Rate |
> |---|:---:|:---:|:---:|:---:|
> | GPQA | 0.473 $\rightarrow$ 0.357 | 24.5 \% | 0.736 $\rightarrow$ 0.651 | 11.5 \% |
> | GSM8K | 0.455 $\rightarrow$ 0.333 | 26.8 \% | 0.727 $\rightarrow$ 0.727 | 0.0 \% |
> | HellaSwag | 0.630 $\rightarrow$ 0.500 | 20.6 \% | 0.739 $\rightarrow$ 0.543 | 26.5 \% |
> | MMLU (Pro. Medicine) | 0.750 $\rightarrow$ 0.268 | 64.3 \% | 0.839 $\rightarrow$ 0.714 | 14.9 \% |
>
>
> | Qwen-7B | Subversion Change | Subversion Drop Rate | Correction Change | Correction Drop Rate |
> |---|:---:|:---:|:---:|:---:|
> | GPQA | 0.717 $\rightarrow$ 0.500 | 30.3 \% | 0.711 $\rightarrow$ 0.553 | 22.2 \% |
> | GSM8K | 0.342 $\rightarrow$ 0.233 | 31.9 \% | 0.740 $\rightarrow$ 0.575 | 22.3 \% |
> | HellaSwag | 0.709 $\rightarrow$ 0.430 | 39.4 \% | 0.767 $\rightarrow$ 0.488 | 36.4 \% |
> | MMLU (Pro. Medicine) | 0.579 $\rightarrow$ 0.389 | 32.8 \% | 0.853 $\rightarrow$ 0.632 | 25.9 \% |

---

> ### Author Response · Authors · 2025-11-20
>
> `A3` *$\Delta$ value calibration*
>
> Thank you for your insightful question! The quality of the belief difference, $\Delta$, indeed warrants further analysis. Below, we (1) describe how we estimate the DCM parameters and verify that the resulting Conformity/Obstinacy/$\Delta$ estimates are well calibrated against empirical "Ground Truth", and (2) explain how we can further calibrate the Identity Bias Coefficient (IBC) to enable more accurate cross-setting comparison of identity bias.
>
>
> **1. DCM parameter estimation**
>
> Using the generated responses, we fit a Dirichlet–Compound–Multinomial (DCM) model to estimate the parameters $\alpha$ and the identity weights $w_i, w_j$, from which Conformity, Obstinacy, and $\Delta$ are derived. We then compare these model-based estimates with Ground Truth values computed directly from the debate trajectories (i.e., empirical transition frequencies) in both anonymized and non-anonymized settings.
>
> As shown in the tables below, the estimated Conformity, Obstinacy, and $\Delta$ closely match their Ground Truth counterparts across models and conditions:
>
>
> | **Qwen7B-GPQA** | Ground Truth | Estimation | Ground Truth (Anonymized) | Estimation (Anonymized) |
> |:---------------:|:-----------:|:----------:|:------------------------:|:-----------------------:|
> |    Conformity   |    0.647    |    0.719   |           0.485          |          0.521          |
> |    Obstinacy    |    0.255    |    0.236   |           0.424          |          0.440          |
> |      $\Delta$     |    0.392    |    0.483   |           0.061          |          0.081          |
>
> | **Qwen7B-MMLU (Pro. Med.)** | Ground Truth | Estimation | Ground Truth (Anonymized) | Estimation (Anonymized) |
> |:---------------------------:|:-----------:|:----------:|:------------------------:|:-----------------------:|
> |          Conformity         |    0.709    |    0.707   |           0.498          |          0.487          |
> |          Obstinacy          |    0.274    |    0.255   |           0.471          |          0.486          |
> |            $\Delta$           |    0.435    |    0.452   |           0.027          |          0.001          |
>
> | **Llama8B-MMLU (Pro. Med.)** | Ground Truth | Estimation | Ground Truth (Anonymized) | Estimation (Anonymized) |
> |:----------------------------:|:-----------:|:----------:|:------------------------:|:-----------------------:|
> |          Conformity          |    0.543    |    0.580   |           0.392          |          0.406          |
> |           Obstinacy          |    0.392    |    0.409   |           0.549          |          0.580          |
> |            $\Delta$            |    0.151    |    0.171   |          -0.157          |          -0.174         |
>
> **The close agreement between Ground Truth and estimated values indicates that the DCM-based belief-difference term $\Delta$ is well calibrated to the empirical debate dynamics**: when the data exhibit stronger deference to peers (or stronger resistance), the fitted $\Delta$ mirrors this behavior. In particular, $\Delta$ changes sign and magnitude in a way that aligns with whether peers are, on average, more likely to pull agents toward correct or incorrect answers, which is precisely the behavior we intend this term to capture.
>
> **2. Further IBC calibration via concentration scaling**
>
> To make identity bias comparable across models, tasks, and debate configurations, we may further calibrate the IBC by scaling with the estimated Dirichlet concentration. Specifically, using the fitted parameters, we define:
>
> \begin{equation}
> \text{IBC}\_\text{calibrated} = \text{IBC} \times  ||\hat{\alpha}_{i,t}||_1 = \frac{w_j - w_i}{{||{\alpha}\_{i,t}||}\_1} \times {||\hat{\alpha}\_{i,t}||}\_1
> \end{equation}
>
> This calibration (1) removes arbitrary scaling in the prior concentration, so that the magnitude of IBC reflects true behavioral asymmetry, and (2) enables fair comparison of identity bias across tasks and models with different debate dynamics.

---

> ### Author Response · Authors · 2025-11-20
>
> `A4` Correlation between $\Delta$ and proxies of model confidence
>
> Thanks for the interesting question! Since $\Delta$ is defined at the aggregate level and is difficult to extract for each individual sample, we instead investigate whether observable proxies of model confidence differ between Conformity and Obstinacy events in the Anonymized setting. Concretely, we compute the perplexity of an agent's initial response and compare the distributions across the two behavioral categories: Conformity and Obstinacy. That is, we plot box-and-whisker distributions of perplexity for (i) instances where an agent conformed to its peer’s answer after debate, and (ii) instances where it maintained its original answer despite disagreement.
>
> In [this anonymized repo figure](https://anonymous.4open.science/r/ICLR2026-1107/delta_compare.png), the two distributions overlap substantially, suggesting that confidence alone does not fully explain the magnitude of $\Delta$. Nonetheless, we observe a slight upward drift in perplexity for Conformity cases, indicating that agents tend to have lower confidence in their original answers when they ultimately defer to their peers. This empirical pattern aligns with the intended semantics of $\Delta$: content-driven belief updates should occur more often when an agent is uncertain about its own response. This supports the interpretation that $\Delta$ correlates with meaningful aspects of model uncertainty rather than arbitrary fluctuations.
>
>
> ---
>
> `A5` *Impact of anonymizaton when domain-relevant personas are present*
>
> Thank you for this interesting question! To evaluate whether anonymization suppresses beneficial identity cues--such as deference to a domain-relevant expert--we explicitly measured how often agents conform to a designated expert persona before and after anonymization. Concretely, on the MMLU Professional Medicine benchmark with heterogeneous Qwen-7B agents, we compared conformity toward a Doctor persona (intended "expert") and a generic Assistant persona in both the vanilla and anonymized MAD settings:
>
>
> | Peer Persona | Conform Rate (Non-Anonymized) | Conform Rate (Anonymized) | Drop Rate |
> |---|---|---|---|
> | Doctor (Expert) | 0.5217 | 0.3478 | 33.3% |
> | Assistant | 0.6429 | 0.4286 | 33.3% |
>
> We observe two key patterns:
>
>
> - **Anonymization reduces conformity at a similar rate for both personas**. The conform rate drops by 33.3% for both the Doctor and Assistant personas, suggesting that anonymization does not selectively suppress deference to the supposed expert; it uniformly dampens identity-driven conformity.
> - Even without anonymization, **agents do not preferentially defer to the "expert" persona**. In fact, the conform rate is higher toward the Assistant persona than toward the Doctor. A likely explanation is that the two personas had very similar single-agent performance (both around 80% accuracy on this benchmark), so the Doctor persona did not actually provide a clearly superior information source that would justify increased deference. When peer answers are comparably reliable, agents have little incentive to systematically favor one persona as an expert.
>
> > **But will identity-bias prove useful if there exists a true expert among agents?**
>
> Although exposing agent identities could occasionally accelerate consensus--such as when an "expert" agent is present--we caution against relying on this effect. Any potential benefit is outweighed by the risk of identity-driven behavioral biases, including undue deference, subversion, or overconfidence tied to perceived status rather than evidence.
>
> In other words, even if identity cues sometimes provide a shortcut to agreement, they introduce undesirable and unpredictable dynamics that undermine robustness and trustworthiness. Thus, identity-based behavior is neither necessary nor sufficiently reliable to justify its risks. We insist that an agent's decision to defer or stand by its position should be grounded solely in the **content and quality of reasoning, without any reliance on identity information**.

---

### Author Response · Authors · 2025-11-20
**General Response to Reviewers and Area Chair**

Dear Reviewers and Area Chair,

We extend our sincere gratitude for the time and effort you have dedicated to reviewing our manuscript. The thoughtful questions and constructive feedback from all the reviewers have been invaluable in improving our manuscript.

We are grateful for the reviewers' recognition of the *novelty* of our framework, and the  *practicality*, *effectiveness*, and *theoretical rigor* of our method. Below, we summarize the key strengths and concerns raised by the reviewers, along with our responses:

---

### Key Strengths noted by the reviewers
1. A **novel framework with strong theoretical grounding** for explaining identity-driven multi-agent debate dynamics using the Dirichlet–Compound–Multinomial model. (Reviewers quLh, eSo1)
3. The **simplicity and practicality of Response Anonymization**, requiring no retraining, architectural changes, or auxiliary objectives. (Reviewers quLh, u3fx, eSo1)
4. **Demonstrated effectiveness of the proposed approach** in reducing identity bias across diverse multi-agent debate settings. (Reviewers quLh, u3fx)
5. A **well-written manuscript and extensive analyses**, covering multiple model families, benchmarks, and evaluation dimensions. (All Reviewers quLh, u3fx, eSo1, RJjL)

---

### Key concerns and how we addressed them
1. *How anonymization affects the overall accuracy in multi-agent debate*. We included additional experimental results analyzing the rates at which agents correct or subvert their answers. These results show that **anonymization improves trustworthiness by producing a substantially larger reduction in subversion than in correction**--indicating more reliable and trustworthy debate dynamics. (All Reviewers quLh, u3fx, eSo1, RJjL)
2. *Whether identity cues (e.g., experts) can sometimes be beneficial.* We showed that "expert" identity labels **do not necessarily encourage conforming behaviors in both non-anonymized and anonymized settings**. We further emphasized that an agent's decision to defer or maintain its position should be grounded on content and quality of reasoning, without any reliance on identity cues. (Reviewers quLh, eSo1, RJjL)
3. *How the Dirichlet-Compound-Multinomial model can be justified.* We showed that the DCM framework is empirically well-founded by **estimating its parameters** and demonstrating that the resulting Conformity and Obstinacy values closely match the observed ground-truth statistics. Moreover, we illustrate how these **estimated parameters enable principled calibration** of the Identity Bias Coefficient (IBC). (Reviewers quLh, RJjL)

Including the concerns outlined above, we have addressed all reviewer questions in detail in the responses below, and updated the manuscript accordingly (updated parts are in blue font). We hope these clarifications fully resolve the outstanding issues.

---

Thank you again for your time, careful evaluation, and for managing the review process. We appreciate your effort, and we hope this summary is helpful.

Sincerely,

Authors.

---

### Note · Authors · 2025-12-18

**Comment:**

Dear Reviewers and Area Chair,

With due respect, we have decided to withdraw this submission.
We sincerely appreciate the time and effort invested by the reviewers and the Area Chair in evaluating our work. The comments and requests for additional analysis were thoughtful and constructive, and they have substantially helped us improve the clarity and quality of our claims.

Best,

Authors

**Withdrawal Confirmation:**

I have read and agree with the venue's withdrawal policy on behalf of myself and my co-authors.